# ADDP: Learning General Representations for Image Recognition and Generation with Alternating Denoising Diffusion Process

**Changyao Tian**[1*]**, Chenxin Tao**[2,3*†]**, Jifeng Dai**[2,4]**, Hao Li**[1]**, Ziheng Li**[2]**, Lewei Lu**[3]**,**
**Xiaogang Wang**[1,3]**, Hongsheng Li**[1]**, Gao Huang**[2]**, Xizhou Zhu**[2,3⊠]

[1]MMLab, CUHK, [2]Tsinghua University, [3]Sensetime Research, [4]Shanghai AI Laboratory
`tcyhost@link.cuhk.edu.hk`, `{tcx20,liziheng20}@mails.tsinghua.edu.cn`
`luotto@sensetime.com`, `{xgwang,hsli}@ee.cuhk.edu.hk`
`{daijifeng,gaohuang,zhuxizhou}@tsinghua.edu.cn`

## Abstract

Image recognition and generation have long been developed independently of each other. With the recent trend towards general-purpose representation learning, the development of general representations for both recognition and generation tasks is also promoted. However, preliminary attempts mainly focus on generation performance, but are still inferior on recognition tasks. These methods are modeled in the vector-quantized (VQ) space, whereas leading recognition methods use pixels as inputs. Our key insights are twofold: (1) pixels as inputs are crucial for recognition tasks; (2) VQ tokens as reconstruction targets are beneficial for generation tasks. These observations motivate us to propose an Alternating Denoising Diffusion Process (ADDP) that integrates these two spaces within a single representation learning framework. In each denoising step, our method first decodes pixels from previous VQ tokens, then generates new VQ tokens from the decoded pixels. The diffusion process gradually masks out a portion of VQ tokens to construct the training samples. The learned representations can be used to generate diverse high-fidelity images and also demonstrate excellent transfer performance on recognition tasks. Extensive experiments show that our method achieves competitive performance on unconditional generation, ImageNet classification, COCO detection, and ADE20k segmentation. Importantly, our method represents the first successful development of general representations applicable to both generation and dense recognition tasks. Code is released at `https://github.com/ChangyaoTian/ADDP`.

## 1 Introduction

Image recognition and image generation are both fundamental tasks in the field of computer vision (Bao et al., 2022; He et al., 2022; Wei et al., 2022; Liu et al., 2021; Dhariwal & Nichol, 2021; Ho et al., 2020; Donahue et al., 2017). Recognition tasks aim to perceive and understand the visual world, while generation tasks aim to create new visual data for various applications. Modern recognition algorithms have already surpassed human performance on many benchmarks (Liu et al., 2021; He et al., 2016), and current generative models can synthesize diverse high-fidelity images (Rombach et al., 2022; Esser et al., 2021). However, these two fields have long been developed independently of each other. Recent years have witnessed a significant trend towards *general-purpose representation learning*. For recognition tasks, researchers have extensively studied general representations that can be adapted to various downstream tasks (Peng et al., 2022; Dong et al., 2023; He et al., 2022; Chen et al., 2022; Tao et al., 2023). Given this unifying trend, it is natural to expect that representations applicable to both recognition and generation tasks could be developed.

Inspired by this, recent works (Li et al., 2022; Yu et al., 2021; Chen et al., 2020a) attempt to learn general representations for both recognition and generation through a specific generative modeling

---

[*]Equal contribution. ⊠Corresponding authors.
[†]The work is done when Chenxin Tao is an intern at SenseTime Research.

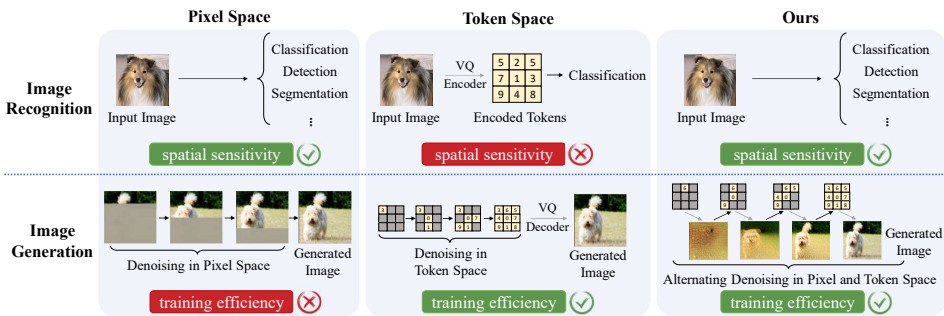

Figure 1: **Inference pipelines of unified methods that learn general representations for both generation and recognition.** Previous methods are modeled either entirely in raw-pixel space (iGPT (Chen et al., 2020a)) or entirely in VQ-token space (ViT-VQGAN (Yu et al., 2021) and MAGE (Li et al., 2022)). In contrast, ADDP exploits both spaces, yielding competitive performances on both recognition and generation tasks.

| Input | #Params | IN-1k Cls $256 \times 256$ | COCO Det $256 \times 256$ | COCO Det $1024 \times 1024$ | ADE20k Seg $256 \times 256$ |
|---|---|---|---|---|---|
| pixel | 86M | 82.6 | 29.4 | 47.5 | 43.7 |
| token | 24M+86M | 81.6 | 12.3 | 31.0 | 31.1 |

Table 1: **Performance of taking pixels or tokens as inputs on canonical recognition tasks.** We adopt a pre-trained VQ tokenizer (Esser et al., 2021) to generate the tokens. Under the same training schedule (see Appendix), using VQ-tokens as inputs is inferior to its pixel counterpart on all tasks, and the gap is even larger on dense recognition tasks.

paradigm, *i.e.*, Masked Image Modeling (MIM) (Bao et al., 2022). As shown in Fig. 1, during the generation process, they iteratively recover the image content for a portion of masked regions. Such generation process has been leveraged for high-fidelity image synthesis (Chang et al., 2022). Meanwhile, each recovery step can be regarded as a special case of MIM using different mask ratios, which has also proved to learn expressive representations for image recognition (He et al., 2022). In particular, ViT-VQGAN (Yu et al., 2021) and MAGE (Li et al., 2022) exhibit remarkable generation performance. Nonetheless, their recognition performances fall short. Specifically, they are still limited to classification task, but are not suitable for dense recognition tasks.

We notice that ViT-VQGAN (Yu et al., 2021) and MAGE (Li et al., 2022), like many image generation methods, are modeled in the vector-quantized (VQ) space (van den Oord et al., 2017). While current SoTA representation learning methods for recognition, such as MAE (He et al., 2022) and BEiT (Bao et al., 2022), all take raw image pixels as inputs. Such observation motivates us to propose the following arguments: (1) *Raw pixels as inputs are crucial for recognition tasks.* Pixels preserve spatially sensitive information better than VQ tokens (Shin et al., 2023), which is particularly useful for dense recognition tasks. As shown in Tab. 1, taking pixels as inputs outperforms the VQ tokens counterpart in typical recognition tasks. (2) *VQ tokens as reconstruction targets are beneficial for generation tasks.* Previous works such as (van den Oord et al., 2017; Rombach et al., 2022) show that compared to generating raw pixels, predicting VQ tokens can help the model eliminate imperceptible image details, mitigating the optimization difficulty and resulting in better image generation quality.

Based on these observations, a natural question arises: *Is it possible to associate the two spaces within a single representation learning framework, allowing the model to perceive in raw pixels and generate in latent visual tokens?*

To this end, we propose a general representation learning framework that bridges pixel and token spaces via an Alternating Denoising Diffusion Process (ADDP). Specifically, at each step in the alternating denoising process, we first decode pixels from previous VQ tokens, and then generate new VQ tokens from these decoded pixels. For the corresponding diffusion process, we first map the original images into VQ-token space with a pre-trained VQ encoder (Chang et al., 2022), then gradually mask out some VQ tokens. An off-the-shelf VQ decoder is employed for the token-to-pixel decoding, while a learnable encoder-decoder network is introduced for the pixel-to-token generation. The training objective is given by the evidence lower bound (ELBO) of the alternating denoising diffusion process. When applied to image generation, we follow the proposed alternating

denoising process to generate images. When applied to image recognition, the learned encoder, which takes raw pixels as inputs, would be fine-tuned on corresponding datasets.

Extensive experiments demonstrate the superior performance of ADDP on image generation and recognition tasks, including unconditional generation on ImageNet $256 \times 256$ (Deng et al., 2009), ImageNet-1k classification, COCO (Lin et al., 2014) detection and ADE20k (Zhou et al., 2019) segmentation. For unconditional generation, ADDP is able to generate high-fidelity images, achieving better performance than previous SoTAs (Li et al., 2022). For recognition tasks, ADDP is competitive with current leading methods tailored for recognition tasks (He et al., 2022; Bao et al., 2022). Specifically, to the best of our knowledge, ADDP is the *first* approach to develop general representations that are applicable to both generation and dense recognition tasks.

## 2 RELATED WORK

Deep generative models are initially developed for image generation. However, recent works found that models trained for some specific *generative tasks*, such as Masked Image Modeling (MIM) (He et al., 2022; Bao et al., 2022), can learn expressive representations that can be transferred to various downstream *recognition tasks*. Such discovery has inspired a series of works that attempt to unify image generation and representation learning.

**Deep Generative Models for Image Generation.** Early attempts (*e.g.*, GANs (Goodfellow et al., 2014; Mirza & Osindero, 2014; Denton et al., 2015; Radford et al., 2016; Chen et al., 2016; Arjovsky et al., 2017; Zhu et al., 2017; Karras et al., 2018; Zhang et al., 2019; Brock et al., 2019), VAEs (Kingma & Welling, 2014; Higgins et al., 2017; Vahdat & Kautz, 2020), and autoregressive models (van den Oord et al., 2016a;b; Salimans et al., 2017)) directly decode raw pixels from random distributions. However, VQ-VAE (van den Oord et al., 2017) points out that directly generating raw pixels is challenging and resource-wasteful due to the redundant low-level information in images. In contrast, VQ-VAE proposes a two-stage paradigm: the first stage encodes images into latent representations (*i.e.*, discrete visual tokens), and the second stage learns to generate visual tokens with powerful autoregressive models. These generated visual tokens are then decoded into raw pixels by the decoder learned in the first stage. Such a two-stage latent space paradigm shows superior training efficiency and performance compared to raw-pixel-wise methods and is thus adopted by most state-of-the-art generative models (Razavi et al., 2019; Yu et al., 2021; Esser et al., 2021; Gu et al., 2022; Chang et al., 2022; Ramesh et al., 2022).

On the other hand, diffusion models (Ho et al., 2020; Dhariwal & Nichol, 2021; Gu et al., 2022; Ramesh et al., 2022; Chang et al., 2022; Saharia et al., 2022; Chang et al., 2023) have also achieved impressive results in image generation, which can produce high-fidelity images by iteratively refining the generated results. For example, Guided Diffusion (Dhariwal & Nichol, 2021) directly decodes raw pixels with diffusion models, and for the first time achieves better results than GAN- and VAE-based generative models. Recent works (*e.g.*, Stable Diffusion (LDMs) (Rombach et al., 2022), VQ-Diffusion (Gu et al., 2022) and MaskGIT (Chang et al., 2022)) further combine diffusion models with the two-stage latent space paradigm, achieving superior image quality. Meanwhile, the success of diffusion models has also been extended to text-to-image generation (Gu et al., 2022; Ramesh et al., 2022; Saharia et al., 2022; Chang et al., 2023), image editing (Zhang et al., 2022; Kawar et al., 2022b), image denoising (Kulikov et al., 2022; Kawar et al., 2022a), etc.

Following previous works, ADDP also performs diffusion with latent space to generate images. The key difference is that our method refines raw pixels and latent representations alternately, which can learn unified representation for both recognition and generation tasks with competitive performance.

**Generative Pre-training for Image Representation Learning.** Recent research (Bao et al., 2022; Peng et al., 2022; Dong et al., 2023; He et al., 2022; Chen et al., 2022; Liu et al., 2022; Wei et al., 2022; Fang et al., 2023) suggests that some specific generative modeling tasks (*e.g.*, Masked Image Modeling (MIM)) can learn more expressive and effective representations than previous representation learning methods (*e.g.*, supervised methods (Dosovitskiy et al., 2021; Liu et al., 2021) and self-supervised discriminative methods (He et al., 2020; Chen et al., 2020c;b)). These generative pre-training methods have shown superior performance when transferred to various downstream recognition tasks, such as image classification, object detection, and semantic segmentation. MIM methods learn representations by reconstructing image content from masked images. For example, BEiTs (Bao et al., 2022; Peng et al., 2022; Dong et al., 2023) reconstruct the discrete visual

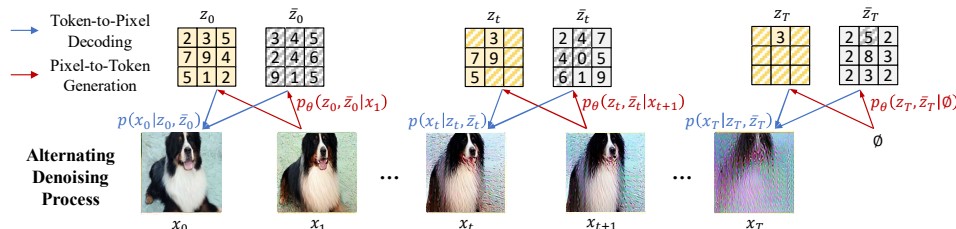

Figure 2: **Alternating denoising process.** Our method first predicts $p_\theta(z_T, \bar{z}_T | \varnothing)$ by directly feeding all `<MASK>` tokens into our decoder $D$ in Eq. (2). At each step $t$, the noisy image $x_t$ is decoded according to Eq. (1), then used to generate new reliable tokens $z_{t-1}$ and unreliable tokens $\bar{z}_{t-1}$ according to Eq. (2). $x_0$ is the final synthesized noisy-free image.

tokens corresponding to masked parts. MAEs (He et al., 2022; Chen et al., 2022) directly reconstruct the masked pixels. Some works (Liu et al., 2022; Wei et al., 2022) also attempt to reconstruct the momentum features of the original images. Apart from these MIM methods, Corrupted Image Modeling (CIM) (Fang et al., 2023) learns representations by reconstructing from corrupted images, which avoids the use of artificial mask tokens that never appear in the downstream fine-tuning stage.

These generative pre-training methods only focus on the representational expressiveness for image recognition, while fail to preserve the quality of reconstructed images and thus unable to be used for image generation tasks directly. In contrast, ADDP learns general representations that perform well for both image recognition and generation tasks.

**Generative Modeling for Unifying Representation Learning and Image Generation.** Early attempts (Donahue et al., 2017; Donahue & Simonyan, 2019) consider representation learning and image generation as dual problems, thus learning two independent networks (*i.e.*, image encoder and image decoder) to solve both tasks at the same time in a dual-learning paradigm. Inspired by generative representation learning (Devlin et al., 2018; Radford et al., 2018; Howard & Ruder, 2018) in NLP fields, iGPT (Chen et al., 2020a) for the first time unifies these two tasks into a single network by learning from autoregressive image generation, providing good performance for both image recognition and image generation tasks. ViT-VQGAN (Yu et al., 2021) further replaces the raw-pixel inputs and outputs with discrete visual tokens for better performance. Recently, MAGE (Li et al., 2022) proposes to replace the autoregressive decoding process with a diffusion method (*i.e.*, MaskGIT (Chang et al., 2022)), following state-of-the-art practices for image generation.

Despite these attempts at unifying representation learning and image generation, their recognition performance is still inferior to state-of-the-art representation learning methods, as they perform representation learning either entirely in raw-pixel space or entirely in latent space. In contrast, ADDP exploits both raw-pixel and latent space, learning general representations that yield competitive performance on both image recognition and image generation tasks.

## 3 METHOD

### 3.1 ALTERNATING DENOISING OF PIXELS AND VQ TOKENS

Previous works (Yu et al., 2021; Li et al., 2022; Dhariwal & Nichol, 2021) mainly perform the denoising process entirely in either continuous raw pixel space or discrete VQ token space. However, given that both raw pixels and VQ tokens are crucial for recognition and generation tasks respectively, we propose to denoise pixels and VQ tokens alternately, as shown in Fig. 2. In each step, we first decode pixels from previously generated VQ tokens, then generate new VQ tokens conditioned on decoded pixels. To associate the two spaces and enable the alternating denoising process, the *token-to-pixel decoding* and *pixel-to-token generation* are introduced as follows.

**Token-to-Pixel Decoding** is widely used in image generation to restore the generated VQ tokens to visual image pixels (van den Oord et al., 2017; Esser et al., 2021; Chang et al., 2022). The VQ decoder subnetworks in off-the-shelf pre-trained VQ tokenizers (*e.g.*, VQGAN (Esser et al., 2021)) could be directly used to perform such decoding. However, existing VQ decoders can only decode images from the complete VQ sequences, while unable to take partial VQ tokens as inputs. In contrast, our denoising process only generates a portion of VQ tokens at each step. To resolve this

inconsistency and facilitate the use of off-the-shelf VQ decoders, we propose to pair these reliable tokens with some unreliable ones. Specifically, at step $t$, given partially generated VQ tokens $z_t$, we further sample additional complementary VQ tokens $\bar{z}_t$ so as to form the complete VQ sequences for decoding the pixel image $x_t$. In order to distinguish $z_t$ and $\bar{z}_t$, we term them as *reliable tokens* and *unreliable tokens* respectively. Note that $\bar{z}_t$ will only be used for decoding the pixel image and not kept to the next step. Then, the conditional probability of $p(x_t|z_t, \bar{z}_t)$ is defined as

$$p(x_t|z_t, \bar{z}_t) = \delta\Big[x_t = \text{VQ-Decoder}\big(z_t \odot (1 - m_t) + \bar{z}_t \odot m_t\big)\Big], \qquad (1)$$

where $\delta$ denotes the Dirac delta distribution, $\odot$ is the element-wise product. $m_t$ is a binary mask indicating the unreliable regions derived from $z_t$. $m_t$ shares the same spatial size with $z_t$ and $\bar{z}_t$, where the regions with binary values of 1 are unreliable. Both the reliable VQ tokens $z_t$ and unreliable VQ tokens $\bar{z}_t$ would be predicted by our models, which will be discussed in Sec. 3.3.

**Pixel-to-Token Generation** has been shown to be effective for recognition tasks in recent representation learning works (Bao et al., 2022; Peng et al., 2022; Dong et al., 2023). Our method introduces a learnable encoder-decoder network for predicting VQ tokens from noisy images to enable representation learning from pixels. As shown in Fig. 4, taking the previously decoded noisy image $x_t$ as inputs, the encoder $E$ first extracts representation from $x_t$, and then the unreliable regions (*i.e.*, $m_t = 1$) of the extracted representation $E(x_t)$ would be replaced with learnable <MASK> token embedding before feeding into the decoder $D$. The decoder will predict $z_{t-1}$ and $\bar{z}_{t-1}$ of the next step based on these inputs. Given the noisy image $x_t$, the conditional probability of generated reliable VQ tokens $z_{t-1}$ and unreliable VQ tokens $\bar{z}_{t-1}$ at the next step $t - 1$ are given as

$$p_\theta(z_{t-1}, \bar{z}_{t-1}|x_t) = D\Big(E(x_t) \odot (1 - m_t) + e_{\text{mask}} \odot m_t\Big), \qquad (2)$$

where $m_t$ is the same binary mask as in Eq. (1), indicating the unreliable regions at step $t$. $E$ and $D$ are learnable encoder and decoder subnetworks respectively. $e_{\text{mask}}$ is a learnable <MASK> token embedding and $\theta$ represents the parameter of the whole network. Since our network takes full images as the inputs, it can adapt various image backbones (*e.g.*, CNNs (He et al., 2016) and ViTs (Dosovitskiy et al., 2021)) as the encoder naturally. Experiments in Sec. 4.2 show that the learned representations from the encoder generalize well for both recognition and generation tasks.

**Alternating Denoising Process** is shown in Fig. 2. Starting from an empty sequence with the unreliable mask $m_{T+1}$ of all 1s, $p_\theta(z_T, \bar{z}_T|\varnothing)$ is predicted by feeding all <MASK> tokens into our decoder $D$ in Eq. (2). After that, at each step $t$, the noisy images $x_t$ are decoded according to Eq. (1) and then used to generate new reliable tokens $z_{t-1}$ and unreliable tokens $\bar{z}_{t-1}$ according to Eq. (2). Finally, the synthesized noisy-free images are the refined images $x_0$ at step 0. The joint distribution of all variables in the alternating denoising process is defined as

$$p_\theta(z_{0:T}, \bar{z}_{0:T}, x_{0:T}) = \underbrace{p_\theta(z_T, \bar{z}_T|\varnothing)}_{\text{our model}} \underbrace{p(x_0|z_0)}_{\text{VQ-Decoder}} \prod_{t=1}^{T} \underbrace{p(x_t|z_t, \bar{z}_t)}_{\text{VQ-Decoder}} \underbrace{p_\theta(z_{t-1}, \bar{z}_{t-1}|x_t)}_{\text{our model}}. \qquad (3)$$

### 3.2 DIFFUSION PROCESS

Following previous denoising diffusion training paradigms (Ho et al., 2020; Gu et al., 2022), we propose a corresponding diffusion process for ADDP as well, as is shown in Fig. 3. Given an image $x_0$, an off-the-shelf pre-trained VQ encoder is used to map $x_0$ to its corresponding VQ

Figure 3: **Diffusion process.**

tokens $z_0 = \text{VQ-Encoder}(x_0)$. Then, the diffusion process gradually masks out some regions of $z_0$ with a Markov chain $q(z_t|z_{t-1})$. As mentioned in Sec. 3.1, $z_{t-1}$ corresponds to the reliable VQ tokens at step $t - 1$, and reliable VQ tokens $z_t$ at step $t$ is a portion of $z_{t-1}$. The whole process consists of $T + 1$ steps in total, where all tokens would be masked out at last.

For the unreliable VQ tokens $\bar{z}_{t-1}$, a forward process $q(\bar{z}_{t-1}|z_t)$ is designed to produce $\bar{z}_{t-1}$ from $z_t$. Instead of $z_{t-1}$, we use the reliable tokens $z_t$ as the condition. Empirical results in Tab. 3 demonstrate the advantage of using $q(\bar{z}_{t-1}|z_t)$ over $q(\bar{z}_{t-1}|z_{t-1})$.

The conditional distribution $q(\bar{z}_{t-1}|z_t)$ is obtained with a token predictor, which receives reliable tokens $z_t$ as inputs and predicts unreliable tokens $\bar{z}_{t-1}$. Since our goal is to generate the original image with unreliable tokens, the optimal value of $q(\bar{z}_{t-1}|z_t)$ should be $q(z_0|z_t)$. However, $q(z_0|z_t)$ is generally intractable, so the token predictor needs to learn from data samples to approximate $q(z_0|z_t)$. To achieve this, it is trained to predict all tokens $z_0$ from reliable tokens $z_t$ only. Finally, as our model is trained to predict unreliable tokens (see Sec. 3.3), the token predictor is only required during training. Therefore, the whole diffusion process is defined as

$$q(z_{0:T}, \bar{z}_{0:T}|x_0) = \underbrace{q(z_0|x_0)}_{\text{VQ-Encoder}} \underbrace{q(\bar{z}_T|\varnothing)}_{\text{token predictor}} \prod_{t=1}^{T} \underbrace{q(z_t|z_{t-1})}_{\text{add mask}} \underbrace{q(\bar{z}_{t-1}|z_t)}_{\text{token predictor}}. \tag{4}$$

For simplicity, the decoded noisy images $x_{1:T}$ are omitted, since the derivation of the training objective (refer to Appendix) shows that they are unnecessary to be included in the diffusion process.

## 3.3 LEARNING THE DENOISING PROCESS

Given the proposed alternating denoising diffusion process, ADDP can be optimized through minimizing the evidence lower bound (ELBO) of $\log p_\theta(x_0)$, which consists of the following terms:

$$L_{\text{ELBO}} = L_{\text{VQ}} + \sum_{t=1}^{T} L_t + L_{T+1}, \tag{5}$$

$$L_{\text{VQ}} = \mathbb{E}_{q(z_0, \bar{z}_0, x_0)}\Big[ -\log \underbrace{p(x_0|z_0)}_{\text{VQ-Decoder}} + \log \underbrace{q(z_0|x_0)}_{\text{VQ-Encoder}} \Big],$$

$$L_t = \mathbb{E}_{q(z_t, \bar{z}_t, z_0)}\Big[ D_{\text{KL}}\Big( q(z_{t-1}|z_t, z_0) \underbrace{q(\bar{z}_{t-1}|z_t)}_{\text{token predictor}} \| \underbrace{p_\theta(z_{t-1}, \bar{z}_{t-1}|x_t = \text{VQ-Decoder}(z_t, \bar{z}_t))}_{\text{our model}} \Big) \Big],$$

$$L_{T+1} = \mathbb{E}_{q(z_0)}\Big[ D_{\text{KL}}\Big( q(z_T|z_0) \underbrace{q(\bar{z}_T|\varnothing)}_{\text{token predictor}} \| \underbrace{p_\theta(z_T, \bar{z}_T|\varnothing)}_{\text{our model}} \Big) \Big],$$

where $D_{\text{KL}}$ is KL divergence. $L_{\text{VQ}}$ corresponds to the training of VQ tokenizer, which is omitted in our training because a pre-trained VQ tokenizer is used. $L_{1:T+1}$ are used to optimize our model parameters $\theta$. $p_\theta$ is given by Eq. (2). Please see Appendix for detailed derivation.

**Training Target.** Like previous MIM methods (He et al., 2022; Bao et al., 2022), we only optimize the loss $L_t$ on masked tokens (*i.e.*, unreliable regions with $m_t = 1$). Following the reparameterization trick in VQ-Diffusion (Gu et al., 2022), $L_t$ can be further simplified as

$$L_t = \mathbb{E}_{q(z_t, \bar{z}_t)}\Big[ D_{\text{KL}}\Big( q(z_0|z_t) \underbrace{q(\bar{z}_{t-1}|z_t)}_{\text{token predictor}} \| \underbrace{p_\theta(z_0, \bar{z}_{t-1}|x_t = \text{VQ-Decoder}(z_t, \bar{z}_t))}_{\text{our model}} \Big) \Big], \tag{6}$$

which implies that $q(z_0|z_t)$ and $q(\bar{z}_{t-1}|z_t)$ are the training targets of our model. Since $q(z_0|z_t)$ is generally intractable, the token predictor $q(\bar{z}_{t-1}|z_t)$ is used as an estimation of $q(z_0|z_t)$ as aforementioned in Sec. 3.2. Therefore, it is feasible to predict $q(\bar{z}_{t-1}|z_t)$ only.

**Training Process.** As shown in Fig. 4, given an image $x_0$, we first compute the reliable tokens $z_t$ and unreliable tokens $\bar{z}_t$ following the diffusion process in Sec. 3.2. $z_t$ and $\bar{z}_t$ are then combined and fed into the VQ decoder to generate the synthesized image $x_t$. After that, the pixel-to-token generation network takes $x_t$ as input and predicts the distribution of $\bar{z}_{t-1}$. The training target $q(\bar{z}_{t-1}|z_t)$ can be computed from the token predictor.

Our model takes the noisy synthesized image $x_t = \text{VQ-Decoder}(z_t, \bar{z}_t)$ as training inputs. Inappropriate values of unreliable $\bar{z}_t$ may hurt the quality of synthesized images, degenerating the performance of image generation and image recognition tasks. To alleviate this issue, instead of directly using $\bar{z}_t$ sampled from $q(\bar{z}_t|z_{t+1})$, a mapping function $f(q(\bar{z}_t|z_{t+1}))$ is introduced for the unreliable parts in Eq. (1) as

$$x_t = \text{VQ-Decoder}\big(z_t \odot (1 - m_t) + f(q(\bar{z}_t|z_{t+1})) \odot m_t\big). \tag{7}$$

Candidates for the mapping function $f(\cdot)$ are designed as {Sampling, ArgMax, WeightedSum}. Sampling is the naïve design that sample tokens according to $q(\bar{z}_t|z_{t+1})$. ArgMax is to choose the tokens with the largest probability in $q(\bar{z}_t|z_{t+1})$. WeightedSum indicates that all VQ embeddings

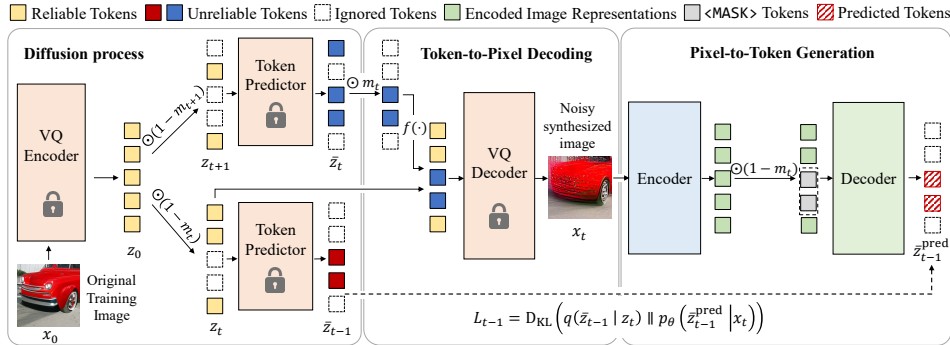

Figure 4: **Training pipeline of ADDP.** The original training image $x_0$ is first encoded into VQ token $z_0$, then a certain timestep $t$ is sampled. The reliable and unreliable tokens $z_t$ and $\bar{z}_t$ are generated according to the diffusion process in Sec. 3.2. After that, $x_t$ is decoded by token-to-pixel decoding in Sec. 3.1. Our pixel-to-token generation network takes $x_t$ as input and generate the prediction of $\bar{z}_{t-1}$. $q(\bar{z}_{t-1}|z_t)$ is used as the training target as mentioned in Sec. 3.3. The lock symbol means that these networks are freezed during training.

within the token codebook would be weighted summed according to $q(\bar{z}_t|z_{t+1})$ before fed into the VQ decoder network. Empirically, we find that using $\mathrm{WeightedSum}$ mapping produces high-quality images and helps to improve our model performance (see Sec. 4.3).

In practice, we also find that feeding the reliable tokens $z_t$ into our decoder $D$ as additional information benefits both recognition and generation tasks, as predicting VQ tokens only from images with considerable noise is difficult when the mask ratio is relatively high.

**Apply to Image Generation.** For image generation, we follow the denoising process mentioned in Sec. 3.1 and Fig. 5 (a). Starting from an empty sequence, our model predicts $z_T$ and $\bar{z}_T$ from pure `<MASK>` token embeddings. At each step, the VQ decoder generates $x_{t+1}$ from tokens of current step $z_{t+1}$ and $\bar{z}_{t+1}$. Then, the pixels $x_{t+1}$ are fed into our encoder-decoder network to predict $z_t$ and $\bar{z}_t$ of the next step. $x_0$ is the final generated noisy-free image.

**Apply to Image Recognition.** As shown in Fig. 5 (b), ADDP can be applied to various recognition tasks after pre-training as well. The encoder takes raw pixels as inputs directly, while the output representations are then forwarded to different task-specific heads. Sec. 4.2 shows that ADDP delivers strong performances after fine-tuning on corresponding datasets.

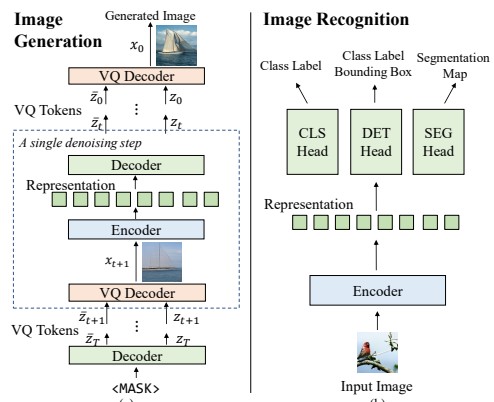

Figure 5: **Inference pipeline of ADDP for image generation and recognition.**

## 4 EXPERIMENTS

### 4.1 IMPLEMENTATION DETAILS

**Network.** The VQ tokenizer is adopted from the off-the-shelf VQGAN (Esser et al., 2021; Chang et al., 2022) model released by MAGE (Li et al., 2022). MAGE-Large is used as the token predictor. Note that the token predictor is used *for training only* and can be discarded in inference. ViT-Large (Dosovitskiy et al., 2021) is adopted as the encoder of ADDP, and the decoder consists of 8 Transformer (Vaswani et al., 2017) blocks.

**Training.** ADDP is trained on ImageNet-1k (Deng et al., 2009) dataset. The total denoising step is $T = 100$. For masking strategy, we sample step $t$ from the values of 1, 2, ..., $T$, while ensuring the mask ratio distribution close to that of MAGE (Li et al., 2022) for a fair comparison. AdamW (Loshchilov & Hutter, 2017) optimizer with a peak learning rate of $1.5 \times 10^{-3}$ is used. The model is trained for 800 epochs with 40 warmup epochs and batch size of 4096.

| Method | #Params. Gen. | Uncond. Gen. | | #Params. Rec. | ImageNet | | COCO | | ADE20k |
|---|---|---|---|---|---|---|---|---|---|
| | | FID↓ | IS↑ | | FT↑ | Linear↑ | $AP^{box}$↑ | $AP^{mask}$↑ | mIoU↑ |
| *Designed for Recognition Only* | | | | | | | | | |
| MoCo v3 (Chen et al., 2021) | - | - | - | 304M | 84.1 | 77.6 | 49.3 | 44.0 | 49.1 |
| BEiT (Bao et al., 2022) | - | - | - | 304M | 85.2 | 52.1 | 53.3 | 47.1 | 53.3 |
| MAE (He et al., 2022) | - | - | - | 304M | 85.9 | 75.8 | 55.6 | 49.2 | 53.6 |
| CAE (Chen et al., 2022) | - | - | - | 304M | 86.3 | 78.1 | 54.5 | 47.6 | 54.7 |
| iBOT (Zhou et al., 2022) | - | - | - | 304M | 84.8 | 81.0 | - | - | - |
| *Designed for Generation Only* | | | | | | | | | |
| BigGAN (Donahue & Simonyan, 2019) | ∼70M | 38.6 | 24.7 | - | - | - | - | - | - |
| ADM (Dhariwal & Nichol, 2021) | 554M | 26.2 | 39.7 | - | - | - | - | - | - |
| MaskGIT (Chang et al., 2022) | 203M | 20.7 | 42.1 | - | - | - | - | - | - |
| IC-GAN (Casanova et al., 2021) | ∼77M | 15.6 | 59.0 | - | - | - | - | - | - |
| *Designed for Both Recognition and Generation* | | | | | | | | | |
| iGPT-L (Chen et al., 2020a) | 1362M | - | - | 1362M | 72.6 | 65.2 | - | - | - |
| ViT-VQGAN (Yu et al., 2021) | 650M | - | - | 650M | - | 65.1 | - | - | - |
| MAGE (Li et al., 2022) | 304M+135M | 9.1 | 105.1 | 304M+24M | 83.9 | 78.9 | 31.4* | 27.6* | 43.1* |
| Ours (ADDP) | 304M+135M | 7.6 | 105.1 | 304M | 85.9 | 23.8 | 54.6 | 48.2 | 54.3 |

Table 2: **Comparison of ADDP with different kinds of existing methods on both visual recognition and generation tasks.** We adopt ViT-L (Dosovitskiy et al., 2021) as the backbone and pre-train it for 800 epochs. The FID (Heusel et al., 2017), IS (Salimans et al., 2016) of unconditional image generation (denoted by *Uncond. Gen.*) is evaluated on ImageNet-1k (Deng et al., 2009) 256×256 validation set; The top-1 accuracy of fine-tuning (FT) and linear probing (Linear) is reported on ImageNet-1k (Deng et al., 2009). $AP^{box}$ and $AP^{mask}$ is reported on COCO (Lin et al., 2014) test-dev set. mIoU is reported on ADE20k (Zhou et al., 2019) validation set. *#Params. Gen.* and *#Params. Rec.* denote the total number of parameters for unconditional generation and recognition backbone, respectively. *For detection and segmentation results, we finetune MAGE using the same training setting. The results of ADDP are marked in  gray .

**Image Recognition.** We evaluate the transfer performance of image classification on ImageNet-1k (Deng et al., 2009), object detection on COCO (Lin et al., 2014) and semantic segmentation on ADE20k (Zhou et al., 2019) respectively. The pre-trained encoder is used as backbone and task-specific heads are appended for different tasks.

**Image Generation.** After training, we use iterative decoding as in MaskGIT (Chang et al., 2022) to iteratively fill in masked tokens and generate images. By default, we recover one token per step.

Please refer to Appendix for more implementation details as well as the generated images of ADDP.

## 4.2 MAIN RESULTS

Tab. 2 compares ADDP with previous methods designed for recognition only, generation only, or both recognition and generation tasks.

**Unconditional Generation.** ADDP surpasses the previous SoTA by 1.5 FID, which validates the effectiveness of our proposed alternating denoising process in generating high-quality images.

**Image Classification.** ADDP achieves comparable fine-tuning performance with methods tailored for recognition. Moreover, compared with the previous SoTA (*i.e.*, MAGE) designed for both recognition and generation, ADDP boosts the performance by 2 points. Such result is consistent with the conclusion drawn from Tab. 1, suggesting pixel inputs are more competent with recognition tasks.

**Linear Probing.** We observe an interesting fact that the linear probing accuracy of ADDP is not aligned with its performance on other tasks, which we assume is caused by the gap between natural images and the *noisy synthetic images* ADDP takes as training inputs. Visualizations of such noisy synthetic images can be found in appendix. Nevertheless, as highlighted in prior studies (He et al., 2022; Bao et al., 2022), linear probing and fine-tuning results are largely uncorrelated, while the core capability of deep neural networks resides in learning strong *non-linear* representations. Therefore, our primary emphasis in this work mainly focus on fine-tuning tasks as well.

**Object Detection and Semantic Segmentation.** Benefiting from our alternating denoising diffusion process, ADDP can also be transferred to object detection and semantic segmentation and achieve competitive performance on these tasks. To the best of our knowledge, this is the *first* work demonstrating that general representations can be learned for both generation and dense prediction tasks. For comparison, we apply MAGE pre-trained ViT-L to these tasks as well, which is trained under the same setting except that it takes VQ tokens as inputs. The results show that our method surpass MAGE by a large margin.

| Conditional Probability | FID ↓ | IS ↑ | FT ↑ |
|---|---|---|---|
| $q(\bar{z}_{t-1}\|z_{t-1})$ | 80.75 | 13.48 | 83.7 |
| $q(\bar{z}_{t-1}\|z_t)$ | 23.63 | 33.09 | 83.5 |

Table 3: **Ablation on condition for unreliable tokens.**

| Mapping Function $f(\cdot)$ | FID ↓ | IS ↑ | FT ↑ |
|---|---|---|---|
| Sampling | 38.79 | 23.44 | 83.2 |
| ArgMax | 26.62 | 30.03 | 83.5 |
| WeightedSum | 23.63 | 33.09 | 83.5 |

Table 4: **Ablation on mapping function $f(\cdot)$.**

| Token Input | FID ↓ | IS ↑ | FT ↑ |
|---|---|---|---|
| encoder & decoder | 20.40 | 37.20 | 81.5 |
| decoder-only | 23.63 | 33.09 | 83.5 |
| None | 36.24 | 21.71 | 83.1 |

Table 5: **Ablation on token input strategy.**

| Prediction Target | Masking Strategy | FID ↓ | IS ↑ | FT ↑ |
|---|---|---|---|---|
| $q(\bar{z}_{t-1}\|z_t)$ | default | 23.63 | 33.09 | 83.5 |
| $\delta(\hat{z}_0 = z_0)$ | default | 39.41 | 18.62 | 83.2 |
| $\delta(\hat{z}_0 = z_0)$ & $q(\bar{z}_{t-1}\|z_t)$ | default | 30.43 | 23.40 | 83.5 |
| $\delta(\hat{z}_0 = z_0)$ | fixed (50%) | 166.54 | 4.73 | 83.4 |

Table 6: **Ablation on prediction target and masking ratio.**

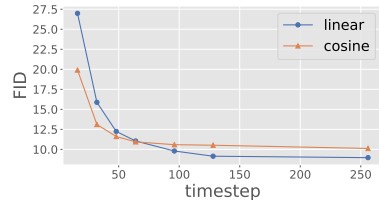

Figure 6: **Ablation on masking schedule for generation.**

## 4.3 Ablation Study

**Ablation Settings.** For ablation, we use ViT-Base as encoder and train ADDP for 150 epochs. The FID (Heusel et al., 2017) and IS (Salimans et al., 2016) scores of unconditional generation with 20 inference steps are reported as well as the fine-tuning accuracy (FT) on ImageNet-1k (Deng et al., 2009). Default settings are marked in gray.

**Probability Condition for Unreliable Tokens.** As in Sec. 3.2, the prediction of $\bar{z}_{t-1}$ is conditioned on $z_t$ rather than $z_{t-1}$. Tab. 3 verifies that using $z_{t-1}$ as condition leads to poor performance.

**Mapping Function for Unreliable Tokens.** Proper mapping function choices can help improve the quality of the learned representation. Tab. 4 shows that directly $\mathrm{Sampling}$ is inferior in both generation and classification, while $\mathrm{WeightedSum}$ can deliver sound performance.

**Token Input.** Directly recovering tokens from pixels with considerable noise may be challenging when the mask ratio is extremely high. Tab. 5 analyzes the effect of feeding tokens to our model, showing that feeding tokens can help enhance the generation ability. However, if tokens are fed directly through the encoder, the classification performance degrades rapidly.

**Prediction Target.** Sec. 3.3 discusses two possible training targets: $q(z_0|z_t)$ and $q(\bar{z}_{t-1}|z_t)$. While $q(z_0|z_t)$ can not be computed directly, we instead sample $z_0$ as an estimated target. Results in Tab. 6 show that predicting $q(\bar{z}_{t-1}|z_t)$ is better on both generation and classification tasks.

**Inference Strategy for Image Generation.** Fig. 6 studies the effect of different inference strategies. Previous works (Chang et al., 2022; Li et al., 2022) adopt cosine masking schedule in inference. However, the generation quickly get saturated when inference steps increase. When using linear masking schedule in inference, similar consistent gain can also be observed as $T$ becomes larger.

## 5 Conclusions

In this paper, we introduce ADDP, a general representation learning framework that is applicable to both image generation and recognition tasks. Our key insights are twofold: (1) pixels as inputs are crucial for recognition tasks; (2) VQ tokens as reconstruction targets are beneficial for generation tasks. To meet the demands of both fields, we propose an Alternating Diffusion Denoising Process (ADDP) to bridge pixel and token spaces. The network is trained to optimize the evidence lower bound (ELBO). Extensive experiments demonstrate its superiority in both image generation and recognition tasks. ADDP for the first time demonstrates that general representation can be learned for both generation and dense recognition tasks.

**Limitations.** ADDP currently relies on a pre-trained VQ Encoder-Decoder, which may constrain generation diversity. Future directions may also include the integration of continuous diffusion processes and scaling to higher resolutions.

**Reproducibility Statement.** The pseudo code of pre-training and unconditional generation can be found in Appendix. A for better understanding. Additionally, the implementation details of our method are fully discussed in Sec. 4.1 and Appendix. D, including both pre-training and various downstream tasks. For the theoretical part, the detailed derivation of ADDP is presented in Appendix. B. The source code and checkpoints are also released at `https://github.com/ChangyaoTian/ADDP`.

**Acknowledgements.** The work is partially supported by the National Key R&D Program of China under Grants NO. 2022ZD0161300 and 2022ZD0114900, by the National Natural Science Foundation of China under Grants 62376134, 62022048 and 62276150, and the Guoqiang Institute of Tsinghua University.

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

## A  PSUEDO CODE

The whole training and inference algorithms are shown in Alg. 1 and Alg. 2. Here discrete-truncnorm denotes the probability distribution used in Sec. 4.1, and $D_{\text{train}}$ denotes the whole training dataset.

### A.1  PRE-TRAINING

---
**Algorithm 1** Pre-training
---
1: **repeat**
2:     sample $t \sim$ discrete-truncnorm$(\{1, \ldots, T\})$
3:     sample $m_t, m_{t+1}$ randomly
4:     sample $x_0 \sim D_{\text{train}}$
5:     $z_0 \leftarrow$ VQ-Encoder$(x_0)$
6:     $z_t, z_{t+1} \leftarrow z_0 \odot (1 - m_t), z_0 \odot (1 - m_{t+1})$
7:     $\bar{z}_{t-1}, \bar{z}_t \leftarrow$ Token-Predictor$(z_t)$, Token-Predictor$(z_{t+1})$      $\triangleright q(\bar{z}_{t-1}|z_t), q(\bar{z}_t|z_{t+1})$
8:     $x_t \leftarrow$ VQ-Decoder$(z_t \odot (1 - m_t) + f(\bar{z}_t) \odot m_t)$      $\triangleright p(x_t|z_t, \bar{z}_t),$ Eq. (7)
9:     $e_t \leftarrow$ Encoder$(x_t)$
10:    $\bar{z}_{t-1}^{\text{pred}} \leftarrow$ Decoder$(e_t \odot (1 - m_t) + e_{\text{mask}} \odot m_t)$      $\triangleright p_\theta(z_0, \bar{z}_{t-1}|x_t),$ Eq. (2)
11:    $L_t \leftarrow$ CE$(\bar{z}_{t-1}, \bar{z}_{t-1}^{\text{pred}}) \odot m_t$      $\triangleright D_{KL},$ Eq. (6)
12:    Take gradient descent step on $\nabla_\theta L_t$
13: **until** converged
---

### A.2  UNCONDITIONAL GENERATION

---
**Algorithm 2** Unconditional Generation
---
1: $z_T, \bar{z}_T \leftarrow$ Decoder$(\varnothing)$      $\triangleright p_\theta(z_T, \bar{z}_T|\varnothing),$ Eq. (2)
2: $x_T \leftarrow$ VQ-Decoder$(z_T \odot (1 - m_T) + f(\bar{z}_T) \odot m_T)$      $\triangleright p(x_T|z_T, \bar{z}_T),$ Eq. (7)
3: **for** $t = T, \ldots, 1$ **do**
4:     $e_t \leftarrow$ Encoder$(x_t)$
5:     $\bar{z}_{t-1}^{\text{pred}} \leftarrow$ Decoder$(e_t \odot (1 - m_t) + e_{\text{mask}} \odot m_t)$      $\triangleright p_\theta(z_{t-1}, \bar{z}_{t-1}|x_t),$ Eq. (2)
6:     sample $z_{t-1}^{\text{pred}} \sim \bar{z}_{t-1}^{\text{pred}}$
7:     $z_{t-1} \leftarrow z_t \odot (1 - m_t) + z_{t-1}^{\text{pred}} \odot (m_t - m_{t-1})$
8:     $\bar{z}_{t-1} \leftarrow f(\bar{z}_{t-1}^{\text{pred}})$
9:     $x_{t-1} \leftarrow$ VQ-Decoder$(z_{t-1} \odot (1 - m_{t-1}) + \bar{z}_{t-1} \odot m_{t-1})$      $\triangleright p(x_{t-1}|z_{t-1}, \bar{z}_{t-1}),$ Eq. (7)
10: **end for**
11: **return** $x_0$
---

## B  DERIVATION FOR ALTERNATING DENOISING DIFFUSION PROCESS

This section presents the detailed derivation for our proposed alternating denoising diffusion process.

**Diffusion Process.** As shown in Sec. 3.2, the diffusion process is described as

$$q(z_{0:T}, \bar{z}_{0:T}|x_0) = \underbrace{q(z_0|x_0)}_{\text{VQ-Encoder}} \underbrace{q(\bar{z}_T|\varnothing)}_{\text{token predictor}} \cdot \prod_{t=0}^{T-1} \underbrace{q(z_{t+1}|z_t)}_{\text{add mask}} \underbrace{q(\bar{z}_t|z_{t+1})}_{\text{token predictor}}, \tag{8}$$

where $x_0$ is the given image, $z_{0:T}$ is the sequence of reliable VQ tokens and $\bar{z}_{0:T}$ is the sequence of unreliable VQ tokens. The diffusion process contains $T + 1$ steps, and all tokens will be masked out at step $T + 1$, resulting in $\varnothing$. For simplicity, we here shift the subscripts $t$ one position to the right (*i.e.* $t - 1 \rightarrow t$).

According to Bayes' Theorem, we have

$$q(z_{t+1}|z_t) = q(z_t|z_{t+1}, z_0) \frac{q(z_{t+1}|z_0)}{q(z_t|z_0)}. \tag{9}$$

Substituting Eq. 9 into Eq. 8 gives

$$q(z_{0:T}, \bar{z}_{0:T}|x_0) = \underbrace{q(z_0|x_0)}_{\text{VQ-Encoder}} \underbrace{q(\bar{z}_T|\varnothing)}_{\text{token predictor}} q(z_T|z_0) \cdot \prod_{t=0}^{T-1} q(z_t|z_{t+1}, z_0) \underbrace{q(\bar{z}_t|z_{t+1})}_{\text{token predictor}}, \quad (10)$$

**Alternating Denoising Process.** On the other hand, Sec 3.1 shows that the alternating denoising process is described as

$$p_\theta(z_{0:T}, \bar{z}_{0:T}, x_{0:T}) = \underbrace{p_\theta(z_T, \bar{z}_T|\varnothing)}_{\text{our model}} \underbrace{p(x_0|z_0)}_{\text{VQ-Decoder}} \cdot \prod_{t=0}^{T-1} \underbrace{p(x_{t+1}|z_{t+1}, \bar{z}_{t+1})}_{\text{VQ-Decoder}} \underbrace{p_\theta(z_t, \bar{z}_t|x_{t+1})}_{\text{our model}}, \quad (11)$$

where $x_{0:T}$ refers to the sequence of decoded images during denoising.

**Evidence Lower Bound.** We can maximize the evidence lower bound (ELBO) as the training objective for the alternating denoising diffusion process. The cross entropy between generated image distribution $p_\theta(x_0)$ and real image distribution $q(x_0)$ is computed as

$$L = -\mathbb{E}_{q(x_0)} \log p_\theta(x_0)$$

$$= -\mathbb{E}_{q(x_0)} \left[ \log \int p_\theta(x_{0:T}, z_{0:T}, \bar{z}_{0:T}) dx_{1:T} dz_{0:T} d\bar{z}_{0:T} \right]$$

$$= -\mathbb{E}_{q(x_0)} \left[ \log \int q(z_{0:T}, \bar{z}_{0:T}|x_0) \frac{p_\theta(x_{0:T}, z_{0:T}, \bar{z}_{0:T})}{q(z_{0:T}, \bar{z}_{0:T}|x_0)} dx_{1:T} dz_{0:T} d\bar{z}_{0:T} \right]$$

$$\leq \underbrace{-\mathbb{E}_{q(z_{0:T}, \bar{z}_{0:T}, x_0)} \log \left[ \frac{\int p_\theta(x_{0:T}, z_{0:T}, \bar{z}_{0:T}) dx_{1:T}}{q(z_{0:T}, \bar{z}_{0:T}|x_0)} \right]}_{\text{using Jensen's inequality}}$$

$$= -\mathbb{E}_{q(z_{0:T}, \bar{z}_{0:T}, x_0)} \left[ \log \frac{p(x_0|z_0)}{q(z_0|x_0)} + \log \frac{p_\theta(z_T, \bar{z}_T|\varnothing)}{q(z_T|z_0)q(\bar{z}_T|\varnothing)} \right.$$

$$\left. + \sum_{t=0}^{T-1} \log \frac{\int p(x_{t+1}|z_{t+1}, \bar{z}_{t+1}) p_\theta(z_t, \bar{z}_t|x_{t+1}) dx_{t+1}}{q(z_t|z_{t+1}, z_0)q(\bar{z}_t|z_{t+1})} \right]$$

$$= -\mathbb{E}_{q(z_{0:T}, \bar{z}_{0:T}, x_0)} \left[ \log \frac{p(x_0|z_0)}{q(z_0|x_0)} + \log \frac{p_\theta(z_T, \bar{z}_T|\varnothing)}{q(z_T|z_0)q(\bar{z}_T|\varnothing)} \right.$$

$$\left. + \sum_{t=0}^{T-1} \log \underbrace{\frac{p_\theta(z_t, \bar{z}_t|x_{t+1} = \text{VQ-Decoder}(z_{t+1}, \bar{z}_{t+1}))}{q(z_t|z_{t+1}, z_0)q(\bar{z}_t|z_{t+1})}}_{p(x_{t+1}|z_{t+1}, \bar{z}_{t+1}) \text{ is deterministic with VQ Decoder}} \right]$$

$$= L_{\text{VQ}} + \sum_{t=0}^{T-1} L_t + L_T, \quad (12)$$

where the inequality holds because of Jensen's inequality. The second last equality holds because we adopt an off-the-shelf VQ decoder to decode pixels from VQ tokens, and such mapping is deterministic. Therefore, the whole objective can be divided into the following terms:

$$L_{\text{ELBO}} = L_{\text{VQ}} + \sum_{t=0}^{T-1} L_t + L_T, \quad (13)$$

$$L_{\text{VQ}} = \mathbb{E}_{q(z_0, \bar{z}_0, x_0)} \left[ -\log \underbrace{p(x_0|z_0)}_{\text{VQ-Decoder}} + \log \underbrace{q(z_0|x_0)}_{\text{VQ-Encoder}} \right],$$

$$L_t = \mathbb{E}_{q(z_{t+1}, \bar{z}_{t+1}, z_0)} \left[ D_{\text{KL}} \left( q(z_t|z_{t+1}, z_0) \underbrace{q(\bar{z}_t|z_{t+1})}_{\text{token predictor}} \| \underbrace{p_\theta(z_t, \bar{z}_t|x_{t+1} = \text{VQ-Decoder}(z_{t+1}, \bar{z}_{t+1}))}_{\text{our model}} \right) \right],$$

$$L_T = \mathbb{E}_{q(z_0)} \left[ D_{\text{KL}} \left( q(z_T|z_0) \underbrace{q(\bar{z}_T|\varnothing)}_{\text{token predictor}} \| \underbrace{p_\theta(z_T, \bar{z}_T|\varnothing)}_{\text{our model}} \right) \right],$$

where $L_{\text{VQ}}$ corresponds to the training of VQ-VAE, and we omit it because we use a pre-trained VQGAN (Esser et al., 2021). $L_{0:T}$ are used to train our model.

**Optimizing the Evidence Lower Bound.** Following the reparameterization trick in VQ-Diffusion (Gu et al., 2022), predicting $q(z_t|z_{t+1}, z_0)$ can be approximated by predicting the noiseless token $z_0$. $L_t$ can thus be written as:

$$
\begin{aligned}
L_t &= \mathbb{E}_{q(z_{t+1}, \bar{z}_{t+1}, z_0)}\left[D_{\text{KL}}\Big(\delta(\hat{z}_0 = z_0)q(\bar{z}_t|z_{t+1}) \,\|\, p_\theta(\hat{z}_0, \bar{z}_t|x_{t+1} = \text{VQ-Decoder}(z_{t+1}, \bar{z}_{t+1}))\Big)\right] \\
&= \mathbb{E}_{q(z_{t+1}, \bar{z}_{t+1})}\mathbb{E}_{q(z_0, \bar{z}_t|z_{t+1}, \bar{z}_{t+1})}\left[\log\frac{q(\bar{z}_t|z_{t+1})}{p_\theta(z_0, \bar{z}_t|x_{t+1} = \text{VQ-Decoder}(z_{t+1}, \bar{z}_{t+1}))}\right] \\
&= \mathbb{E}_{q(z_{t+1}, \bar{z}_{t+1})}\mathbb{E}_{q(z_0, \bar{z}_t|z_{t+1}, \bar{z}_{t+1})}\left[\log\frac{q(z_0|z_{t+1}, \bar{z}_{t+1})q(\bar{z}_t|z_{t+1})}{p_\theta(z_0, \bar{z}_t|x_{t+1} = \text{VQ-Decoder}(z_{t+1}, \bar{z}_{t+1}))}\right] + C_1,
\end{aligned}
\tag{14}
$$

where $C_1 = -\mathbb{E}_{q(z_0, z_{t+1}, \bar{z}_{t+1})}\big[\log q(z_0|z_{t+1}, \bar{z}_{t+1})\big]$ is a constant that can be ignored.

Note that

$$
\begin{aligned}
q(z_0|z_{t+1}, \bar{z}_{t+1}) &= \frac{q(z_0, z_{t+1}, \bar{z}_{t+1})}{q(z_{t+1}, \bar{z}_{t+1})} \\
&= \frac{q(\bar{z}_{t+1}|z_0, z_{t+1})q(z_0, z_{t+1})}{q(\bar{z}_{t+1}|z_{t+1})q(z_{t+1})} \\
&= q(z_0|z_{t+1}),
\end{aligned}
\tag{15}
$$

where $q(\bar{z}_{t+1}|z_0, z_{t+1}) = q(\bar{z}_{t+1}|z_{t+1})$ because the diffusion process is a Markov chain (see Fig. 3). Therefore, we can simplify $L_t$ as

$$
\begin{aligned}
L_t &= \mathbb{E}_{q(z_{t+1}, \bar{z}_{t+1})}\mathbb{E}_{q(z_0, \bar{z}_t|z_{t+1}, \bar{z}_{t+1})}\left[\log\frac{q(z_0|z_{t+1})q(\bar{z}_t|z_{t+1})}{p_\theta(z_0, \bar{z}_t|x_{t+1} = \text{VQ-Decoder}(z_{t+1}, \bar{z}_{t+1}))}\right], \\
&= \mathbb{E}_{q(z_{t+1}, \bar{z}_{t+1})}\left[D_{\text{KL}}\Big(q(z_0|z_{t+1})\underbrace{q(\bar{z}_t|z_{t+1})}_{\text{token predictor}} \,\|\, \underbrace{p_\theta(z_0, \bar{z}_t|x_{t+1} = \text{VQ-Decoder}(z_{t+1}, \bar{z}_{t+1}))}_{\text{our model}}\Big)\right].
\end{aligned}
\tag{16}
$$

Eq. 16 shows that $q(z_0|z_{t+1})$ and $q(\bar{z}_t|z_{t+1})$ are two optimization targets of our model. While $q(z_0|z_{t+1})$ is generally intractable, $q(\bar{z}_t|z_{t+1})$ can serve as a good approximation (see Sec. 3.3). We adopt $q(\bar{z}_t|z_{t+1})$ as the training target in practice. In this way, the loss for use is computed as

$$
\begin{aligned}
L_t &= \mathbb{E}_{q(z_{t+1}, \bar{z}_{t+1})}\left[D_{\text{KL}}\Big(q(\bar{z}_t|z_{t+1}) \,\|\, p_\theta(\bar{z}_t|x_{t+1} = \text{VQ-Decoder}(z_{t+1}, \bar{z}_{t+1}))\Big)\right] \\
&= \mathbb{E}_{q(z_{t+1}, \bar{z}_{t+1})}\left[\text{CE}\Big(\underbrace{q(\bar{z}_t|z_{t+1})}_{\text{token predictor}}, \underbrace{p_\theta(\bar{z}_t|x_{t+1} = \text{VQ-Decoder}(z_{t+1}, \bar{z}_{t+1}))}_{\text{our model}}\Big)\right] - C_2,
\end{aligned}
\tag{17}
$$

where $C_2 = -\mathbb{E}_{q(\bar{z}_t, z_{t+1})}[q(\bar{z}_t|z_{t+1})]$ is a constant, and CE refers to cross entropy.

## C ADDITIONAL EXPERIMENTS

### C.1 RESULTS OF VIT-B

We apply ADDP to ViT-B (Dosovitskiy et al., 2021) and report its performance on image generation and recognition tasks in Tab. 7. The training setting is almost the same as ViT-L, except that we train for 1600 epochs. We use the pre-trained MAGE Base model (Li et al., 2022) as the token predictor. Please refer to Sec. D.2 for implementation details.

**Unconditional Generation.** We obtain $\sim 2$ FID improvement over previous SOTA, demonstrating the powerful generation capacity of ADDP.

| Method | #Params. Gen. | Uncond. Gen. FID↓ | IS↑ | #Params. Rec. | ImageNet FT↑ | Linear↑ | COCO $AP^{box}$↑ | $AP^{mask}$↑ | ADE20k mIoU↑ |
|---|---|---|---|---|---|---|---|---|---|
| *Designed for Recognition Only* | | | | | | | | | |
| MoCo v3 (Chen et al., 2021) | - | - | - | 86M | 83.0 | 76.7 | 47.9 | 42.7 | 47.3 |
| DINO (Caron et al., 2021) | - | - | - | 86M | 83.3 | 77.3 | 46.8 | 41.5 | 47.2 |
| BEiT (Bao et al., 2022) | - | - | - | 86M | 83.2 | 37.6 | 49.8 | 44.4 | 47.1 |
| CIM (Fang et al., 2023) | - | - | - | 86M | 83.3 | - | - | - | 43.5 |
| MAE (He et al., 2022) | - | - | - | 86M | 83.6 | 68.0 | 51.6 | 45.9 | 48.1 |
| CAE (Chen et al., 2022) | - | - | - | 86M | 83.9 | 70.4 | 50.0 | 44.0 | 50.2 |
| iBOT (Zhou et al., 2022) | - | - | - | 86M | 84.0 | 79.5 | 51.2 | 44.2 | 50.0 |
| SiameseIM (Tao et al., 2023) | - | - | - | 86M | 84.1 | 78.0 | 52.1 | 46.2 | 51.1 |
| *Designed for Generation Only* | | | | | | | | | |
| BigGAN (Donahue & Simonyan, 2019) | ∼70M | 38.6 | 24.70 | - | - | - | - | - | - |
| ADM (Dhariwal & Nichol, 2021) | 554M | 26.2 | 39.70 | - | - | - | - | - | - |
| MaskGIT (Chang et al., 2022) | 203M | 20.7 | 42.08 | - | - | - | - | - | - |
| IC-GAN (Casanova et al., 2021) | ∼77M | 15.6 | 59.00 | - | - | - | - | - | - |
| *Designed for Both Recognition and Generation* | | | | | | | | | |
| iGPT-L (Chen et al., 2020a) | 1362M | - | - | 1362M | 72.6 | 65.2 | - | - | - |
| ViT-VQGAN (Yu et al., 2021) | 650M | - | - | 650M | - | 65.1 | - | - | - |
| MAGE (Li et al., 2022) | 86M+90M | 11.0* | 95.42* | 86M+24M | 82.5 | 74.7 | 36.3* | 31.3* | 39.6* |
| Ours (ADDP) | 86M+90M | 8.9 | 95.32 | 86M | 83.9 | 11.5 | 51.7 | 45.8 | 48.1 |

Table 7: **Comparison of ADDP with different kinds of existing methods on both visual recognition and generation tasks.** The FID (Heusel et al., 2017), IS (Salimans et al., 2016) of unconditional image generation (denoted by *Uncond. Gen.*) is evaluated on ImageNet-1k (Deng et al., 2009) 256×256 validation set; The top-1 accuracy of fine-tuning (FT) and linear probing (Linear) is reported on ImageNet-1k (Deng et al., 2009). $AP^{box}$ and $AP^{mask}$ is reported on COCO (Lin et al., 2014) test-dev set. mIoU is reported on ADE20k (Zhou et al., 2019) validation set. *#Params. Gen.* and *#Params. Rec.* denote the total number of parameters for unconditional generation and recognition backbone, respectively.
\* The generation performance of MAGE is re-evaluated using our inference strategy for fair comparison. The original FID and IS scores of MAGE are 11.1 and 81.17, respectively. The detection and segmentation results are run by us using the same training setting. The results of our method are marked in gray .

**Image Classification.** The performance of our method is comparable to those specifically designed for recognition tasks. Similar to our ViT-L model, using ViT-B as the encoder outperforms the previous best model that supports both recognition and generation by 1.4 percentage points. However, we also observe low linear probing performance, which is likely due to the noisy synthetic images as training input, as shown in Fig. 9.

**Object Detection and Semantic Segmentation.** ADDP can achieve comparable performance to methods designed for recognition tasks, suggesting that our model can learn general representations suitable for dense prediction tasks.

## C.2 RESULTS OF RESNET-50

Given that ADDP takes full images as inputs during pre-training, it is architecture-agnostic and thus can be applied to other network structures such as convolution networks. To further demonstrate this, we use ResNet50 (He et al., 2016) as the image encoder and pretrain it for 300 epochs. The performance on generation and recognition tasks are reported in Tab. 8. More implementation details can be found in Tab. 14 and Tab. 17.

**Unconditional Generation.** The unconditional generation performance of our ResNet50 model is comparable to previous methods speciallly designed for generation tasks. To the best of our knowledge, this is the first time that ResNet is used as the image encoder for image generation .

**Image Classification.** Our method's finetuning performance on ImageNet-1k outperforms previous supervised and self-supervised methods using ResNet50 backbone.

## C.3 ROBUSTNESS EVALUATION

We evaluate the robustness of our model in Tab. 9. We use the ImageNet-1k finetuned model from Tab. 7, and run inference on different variants of ImageNet validation datasets (Hendrycks & Di-

| Method | #Params. Gen. | Uncond. Gen. FID↓ | IS↑ | #Params. Rec. | ImageNet FT 100ep↑ | FT 300ep↑ |
|---|---|---|---|---|---|---|
| *Designed for Recognition Only* | | | | | | |
| RSB-A2 (Wightman et al., 2021) | - | - | - | 26M | - | 79.8 |
| RSB-A3 (Wightman et al., 2021) | - | - | - | 26M | 78.1 | - |
| SimSiam† (Chen & He, 2021) | - | - | - | 26M | - | 79.1 |
| MoCo-v2† (Chen et al., 2020c) | - | - | - | 26M | - | 79.6 |
| SimCLR† (Chen et al., 2020b) | - | - | - | 26M | - | 80.0 |
| BYOL† (Grill et al., 2020) | - | - | - | 26M | - | 80.0 |
| SwAV† (Caron et al., 2020) | - | - | - | 26M | - | 80.1 |
| CIM (Fang et al., 2023) | - | - | - | 26M | 78.6 | 80.4 |
| *Designed for Generation Only* | | | | | | |
| BigGAN (Donahue & Simonyan, 2019) | ∼70M | 38.6 | 24.7 | - | - | - |
| ADM (Dhariwal & Nichol, 2021) | 554M | 26.2 | 39.7 | - | - | - |
| MaskGIT (Chang et al., 2022) | 203M | 20.7 | 42.1 | - | - | - |
| IC-GAN (Casanova et al., 2021) | ∼77M | 15.6 | 59.0 | - | - | - |
| *Designed for Both Recognition and Generation* | | | | | | |
| Ours | 26M+90M | 17.1 | 40.1 | 26M | 79.7 | 80.9 |

Table 8: **Comparison of ADDP with different kinds of existing methods on both visual recognition and generation tasks.** We adopt ResNet50 (He et al., 2016) as the backbone and pre-train it for 300 epochs. †indicates the performance results are from CIM (Fang et al., 2023). The results of our method are marked in  gray .

| Method | IN-A top-1 | IN-R top-1 | IN-Sketch top-1 | IN-C 1-mCE | avg |
|---|---|---|---|---|---|
| MSN (Assran et al., 2022) | 37.5 | 50.0 | 36.3 | 53.4 | 44.3 |
| MoCo-v3 (Chen et al., 2021) | 32.4 | 49.8 | 35.9 | 55.4 | 43.4 |
| MAE (He et al., 2022) | 35.9 | 48.3 | 34.5 | 48.3 | 41.8 |
| SiameseIM (Tao et al., 2023) | 43.8 | 52.5 | 38.3 | 57.1 | 47.9 |
| Ours | 35.2 | 54.4 | 40.9 | 57.3 | 47.0 |

Table 9: **Robustness evaluation with ViT-B backbone.**

etterich, 2019; Hendrycks et al., 2021b;a; Wang et al., 2019). The results show that our model can achieve on-par performances with previous best method, *i.e.*, SiameseIM (Tao et al., 2023) that conbines contrastive learning with masked image modeling. We speculate that training with noisy synthetic images may enhance the robustness of our model.

## C.4 RELIABLE TOKEN DETERMINATION MECHANISM

In this section, we study the effect of mechanism to determine the reliable tokens during inference. By default, we generate reliable tokens using an iterative decoding strategy following MaskGIT (Chang et al., 2022) during inference. New reliable tokens are determined based on the predicted probability $p_\theta(\bar{z}_{t-1}|x_t)$ for a given timestep $t$. Specifically, for each masked location, we first sample a candidate token id and its corresponding probability $p$ from the predicted distribution. Then the confidence score $s$ of each location is calculated by adding a Gumbel noise $\epsilon$ to the log probability, i.e. $s = log(p) + \tau\epsilon$. Here, $\tau$ is set to be $6.0 \times \frac{t}{T}$ by default. (please refer to Appendix D.4 for more details). We explore two factors here:

(1) truncating the probability distribution employed for sampling each candidate token by using nucleus sampling (Holtzman et al., 2019), denoted by top-p;

(2) varying the value of $\tau$ to adjust the randomness introduced by the Gumbel noise when computing the confidence scores.

The results in Tab. 10 and Tab. 11 imply that the default setting is optimal, yielding the lowest FID. However, it's noteworthy that the IS score benefits from slightly reducing the value of top-p and $\tau$.

| Method | FID↓ | IS↑ | Precison↑ | Recall↑ |
|---|---|---|---|---|
| ADM (Dhariwal & Nichol, 2021) (w/o classifier guidance) | 26.2 | 39.7 | 0.61 | 0.63 |
| ADM (Dhariwal & Nichol, 2021) (w/ classifer guidance) | 12.0 | 95.4 | 0.76 | 0.44 |
| MAGE (Li et al., 2022) | 9.1 | 105.1 | 0.75 | 0.47 |
| Ours | 7.6 | 105.1 | 0.74 | 0.49 |

Table 12: **More metrics for evaluating both generation quality and diversity of ADDP.**

This suggests that disregarding tokens with excessively low confidence can enhance the quality of synthesized images.

| top-p | FID↓ | IS↑ |
|---|---|---|
| 1.0 | 7.6 | 105.1 |
| 0.95 | 7.9 | 117.3 |
| 0.9 | 10.5 | 124.1 |
| 0.7 | 34.1 | 88.6 |
| 0.5 | 90.9 | 32.9 |
| 0.3 | 185.8 | 8.4 |
| 0.1 | 325.8 | 2.3 |

Table 10: **Effect of truncating the sampling probability distribution.**

| $\tau$ | FID↓ | IS↑ |
|---|---|---|
| 0.0 | 353.7 | 1.3 |
| 2.0 | 18.5 | 107.2 |
| 6.0 | 12.1 | 80.7 |
| 20.0 | 27.3 | 48.7 |
| 60.0 | 33.8 | 41.0 |
| $2.0 \times \frac{t}{T}$ | 23.3 | 109.4 |
| $6.0 \times \frac{t}{T}$ | 7.6 | 105.1 |
| $20.0 \times \frac{t}{T}$ | 17.7 | 63.9 |
| $60.0 \times \frac{t}{T}$ | 27.9 | 48.3 |

Table 11: **Effect of varying the noise coefficient $\tau$.**

## C.5 EVALUATION ON GENERATION DIVERSITY

We evaluate the generation diversity of our model in Tab. 12. Although the non-VQ model ADM (Dhariwal & Nichol, 2021) without classifier guidance achieves high recall, its FID and IS are significantly worse than ours. Besides that, our method achieves both better FID, IS and recall compared to ADM (Dhariwal & Nichol, 2021) with classifier guidance and MAGE (Li et al., 2022) on unconditional generation.

## D IMPLEMENTATION DETAILS

### D.1 COMPARISON OF INPUTS FOR RECOGNITION TASKS

We conduct a preliminary comparison of inputs for recognition tasks in Tab. 1. Since the off-the-shelf VQ tokenizer is trained with resolution $256 \times 256$, we conduct most tasks with input resolution $256 \times 256$ as well. Besides that, we also evaluate the performance on COCO (Lin et al., 2014) detection with resolution $1024 \times 1024$.

**Image Classification.** We train on ImageNet-1k (Deng et al., 2009) from scratch. The training setting mainly follows the one used in MAE (He et al., 2022), except that the input resolution is $256 \times 256$. Detailed hyper-parameters are listed in Tab. 13.

**Object Detection.** We train on COCO (Lin et al., 2014) dataset, with the backbone initialized from the model pre-trained on ImageNet-1k. The training settings are almost the same as in Tab. 18; only the training epoch is changed to 25. We report the results with input resolution $256 \times 256$ and $1024 \times 1024$, respectively.

**Semantic Segmentation.** We train on ADE20k (Zhou et al., 2019) dataset, with the backbone initialized from the model pre-trained on ImageNet-1k. The training settings are almost the same as in Tab. 19; only the input resolution is changed to $256 \times 256$.

| Hyper-parameters | Value |
|---|---|
| Input resolution | $256 \times 256$ |
| Training epochs | 300 |
| Warmup epochs | 20 |
| Batch size | 4096 |
| Optimizer | AdamW |
| Peak learning rate | $1.6 \times 10^{-3}$ |
| Learning rate schedule | cosine |
| Weight decay | 0.3 |
| AdamW $\beta$ | (0.9, 0.95) |
| Erasing prob. | 0.25 |
| Rand augment | 9/0.5 |
| Mixup prob. | 0.8 |
| Cutmix prob. | 1.0 |
| Label smoothing | 0.1 |
| Stochastic depth | 0.1 |

Table 13: **Hyper-parameters for training from scratch on ImageNet.**

### D.2 PRE-TRAINING SETTING

**Network Structure.** The VQ tokenizer used for ADDP is from the off-the-shelf VQGAN (Esser et al., 2021; Chang et al., 2022) model released by MAGE (Li et al., 2022). We also use its ViT-Base model as the token predictor by default. For our encoder-decoder network, we use different models including ViT-B, ViT-L (Dosovitskiy et al., 2021) and ResNet50 (He et al., 2016) as the encoder, while the decoder is composed of 8 Transformer (Vaswani et al., 2017) blocks with 768 feature dimension (or 1024 for ViT-L). In addition, the decoder takes three independent sets of learnable positional embeddings for pixels, VQ-tokens, and <MASK> tokens inputs, respectively.

**Training Setting.** All the models are trained on ImageNet-1k (Deng et al., 2009) dataset. The total denoising step is $T = 100$. The values of sampled mask ratios during training are computed following $\cos(\frac{\pi}{2} \cdot \frac{t}{T})$, where $t = 1, 2, \ldots, T$. The corresponding probability densities for these mask ratios are then calculated from the truncated normal distribution used in MAGE (Li et al., 2022) with mean and standard deviation of 0.55 and 0.25, respectively, truncated probabilities between 0.5 and 1.0. Finally, the mask ratio is sampled based on the normalized discrete probability distribution, as is shown in Fig. 12. To further adapt our model to different denoising timesteps during inference, we sample $\Delta t$ from a uniform discrete distribution from 1 to 5 and replace $z_{t+1}$ with $z_{t+\Delta t}$. Detailed hyper-parameters are listed in Tab. 14.

**The relationship between ADDP and MAGE.** Similar to BEiT (Bao et al., 2022), where the output of dVAE (Ramesh et al., 2021) is used as the training target, we also take the predicted token distribution of MAGE (Li et al., 2022) as part of ADDP's training objective. However, we claim that the two methods are *inherently different* as MAGE is VQ-token based whereas ours is raw-pixel based. Moreover, the results in Tab. 2 and Tab. 7 also demonstrate that ADDP can achieve better performance on both generation and recognition, especially for dense recognition tasks.

### D.3 APPLY TO IMAGE RECOGNITION

We use the pre-trained encoder as the backbone and append task-specific heads for different tasks. We mainly follow the transfer setting in MAE (He et al., 2022).

**Image Classification.** We train on ImageNet-1k (Deng et al., 2009) dataset. The detailed hyper-parameters of finetuning and linear probing for ViT backbone are listed in Tab. 15 and Tab. 16, while the finetuning hyper-parameters for ResNet50 are listed in Tab 17.

**Object Detection.** We train on COCO (Lin et al., 2014) dataset. We follow ViTDet (Li et al., 2022) to use Mask R-CNN (He et al., 2017) as the detection head. The detailed hyper-parameters are listed in Tab. 18.

| Hyper-parameters | Value |
|---|---|
| Input resolution | $256 \times 256$ |
| Training epochs | 1600 (ViT-B) / 800 (ViT-L) 300 (ResNet50) |
| Warmup epochs | 40 (ViT) / 10 (ResNet50) |
| Batch size | 4096 (ViT) / 2048 (ResNet50) |
| Optimizer | AdamW |
| Peak learning rate | $1.5 \times 10^{-3}$ |
| Learning rate schedule | cosine |
| Weight decay | 0.05 |
| AdamW $\beta$ | (0.9, 0.95) (ViT) (0.9, 0.98) (ResNet50) |
| Augmentation | RandomResizedCrop(0.2, 1.0) RandomHorizontalFlip(0.5) |
| Label smoothing | 0.1 |
| Drop out rate | 0.1 |

Table 14: **Hyper-parameters for pre-training.**

| Hyper-parameters | Value |
|---|---|
| Input resolution | $256 \times 256$ |
| Finetuning epochs | 100 (B) / 50 (L) |
| Warmup epochs | 20 |
| Batch size | 1024 |
| Optimizer | AdamW |
| Peak learning rate | $4.0 \times 10^{-3}$ (B) $2.5 \times 10^{-4}$ (L) |
| Learning rate schedule | cosine |
| Weight decay | 0.05 |
| Adam $\beta$ | (0.9, 0.999) |
| Layer-wise learning rate decay | 0.65 (B) / 0.8 (L) |
| Erasing prob. | 0.25 |
| Rand augment | 9/0.5 |
| Mixup prob. | 0.8 |
| Cutmix prob. | 1.0 |
| Label smoothing | 0.1 |
| Stochastic depth | 0.1 |

Table 15: **Hyper-parameters for ImageNet finetuning with ViT backbone.**

**Semantic Segmentation.** We train on ADE20k (Zhou et al., 2019) dataset. We use UperNet (Xiao et al., 2018) as the segmentation head. The detailed hyper-parameters are listed in Tab. 19.

### D.4 APPLY TO IMAGE GENERATION

We adopt an iterative decoding strategy following MaskGIT (Chang et al., 2022) when applying to image generation tasks. Given a blank canvas, the decoder first predicts $z_T$ and $\bar{z}_T$ from pure <MASK> token embeddings. Then the VQ decoder generates the initial image $x_T$ based on $\bar{z}_T$ only. After that, we iteratively decode more reliable tokens $z_t$ and the corresponding image $x_t$ until finally generating the noisy-free image $x_0$.

During the iteration, new reliable tokens $z_t$ for each masked location are sampled based on its prediction probability. The confidence score for each sampled token is its probability plus a Gumbel Noise, of which the temperature $\tau$ is set to be $6.0 \times \frac{t}{T}$ by default. Previously generated reliable tokens $z_{t+1}$ will always be kept by setting its corresponding score to 1.0 manually.

| Hyper-parameters | Value |
|---|---|
| Input resolution | $224 \times 224$ |
| Finetuning epochs | 90 |
| Batch size | 16384 |
| Optimizer | LARS |
| Peak learning rate | 6.4 |
| Learning rate schedule | cosine |
| Warmup epochs | 10 |
| Data augment | RandomResizedCrop(0.08, 1.0) RandomHorizontalFlip(0.5) |

Table 16: **Hyper-parameters for ImageNet linear probing.**

| Hyper-parameters | Value |
|---|---|
| Input resolution | $256 \times 256$ |
| Finetuning epochs | 100 / 300 |
| Warmup epochs | 5 |
| Batch size | 2048 |
| Optimizer | AdamW |
| Peak learning rate | $8.0 \times 10^{-3}$ (100ep) $5.0 \times 10^{-3}$ (300ep) |
| Learning rate schedule | cosine |
| Weight decay | 0.02 |
| Adam $\beta$ | (0.9, 0.999) |
| Layer-wise learning rate decay | None |
| Loss Type | BCE |
| Erasing prob. | None |
| Rand augment | 6/0.5 (100ep) / 7/0.5 (300ep) |
| Repeated Aug | $\times$ (100ep) / $\checkmark$ (300ep) |
| Mixup prob. | 0.1 |
| Cutmix prob. | 1.0 |
| Label smoothing | 0.1 |
| Stochastic depth | None (100ep) / 0.05 (300ep) |

Table 17: **Hyper-parameters for ImageNet finetuning with ResNet50 backbone.**

As for generating the next step's mask $m_{t-1}$, we mask out the last $k$ tokens of $z_t$ based on their prediction scores. Here the exact value of $k$ depends on the masking schedule and the total inference steps $T$. Specifically, we have $k = \cos(\frac{\pi}{2} \cdot \frac{T-t}{T})$ for *cosine* schedule and $k = \frac{t}{T}$ for *linear* schedule.

# E  VISUALIZATION

**Unconditional Generation.** We provide more unconditional image samples generated by ADDP in Fig. 7.

**Intermediate Generated Results.** We also show some intermediate generated results in Fig. 8. Note that the masked image here only indicates the corresponding positions of reliable tokens $z_t$ in each step, whereas in real implementations we feed the entire image into our encoder.

**Synthetic Training Images.** Fig. 9, Fig. 10 and Fig. 11 show some synthetic images generated by different mapping functions as training input. Qualitatively, the $\mathrm{WeightedSum}$ strategy synthesizes images with better quality than its couterparts and thus achieves better performance in both recognition and generation tasks, as is shown in Tab. 4.

**Distribution of Mask Ratio and Timestep for Pre-training.** Fig. 12 shows the discrete distribution of mask ratio and timesteps during pre-training respectively.

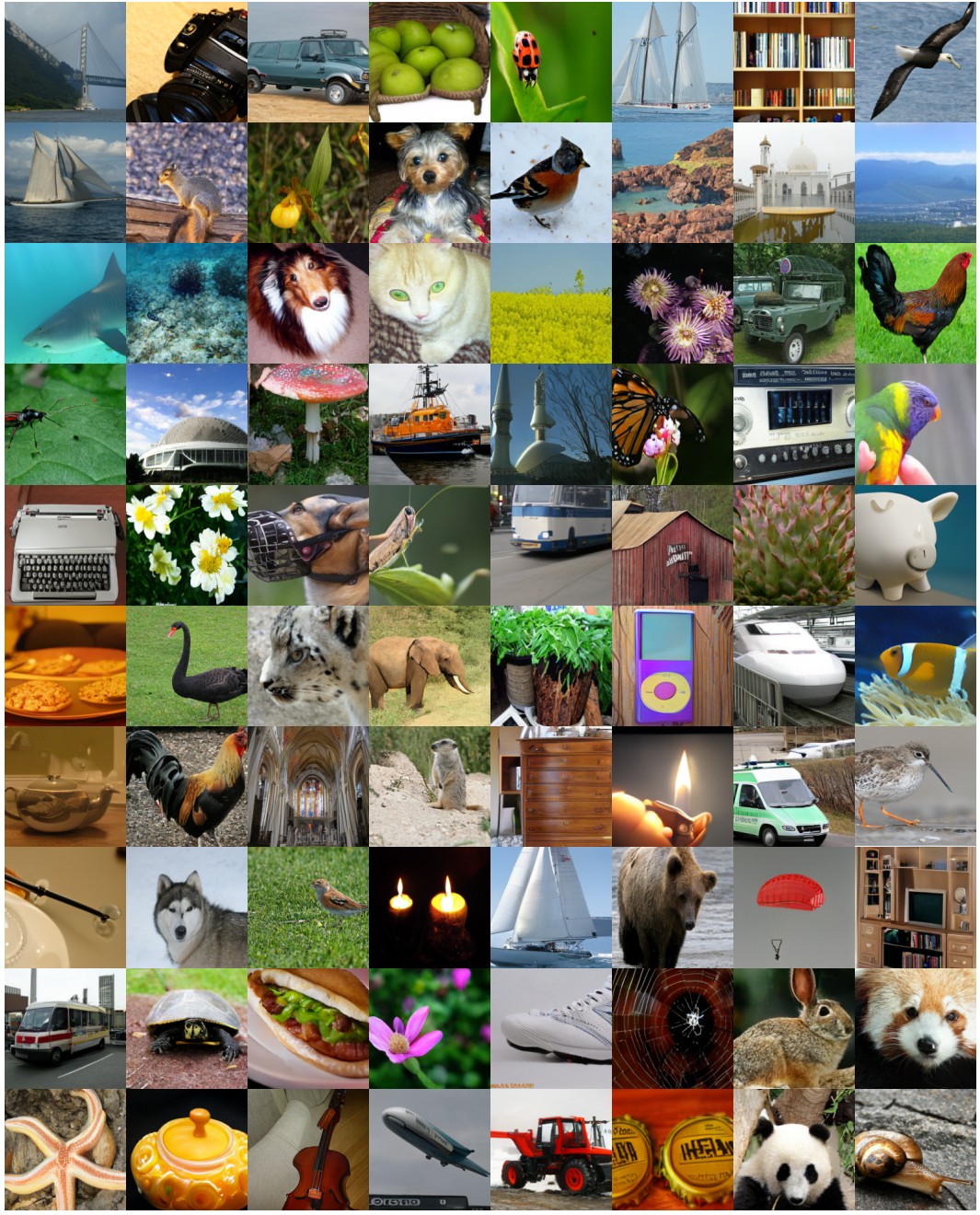

Figure 7: **Visualizations of unconditional generated images on ImageNet-1k.**

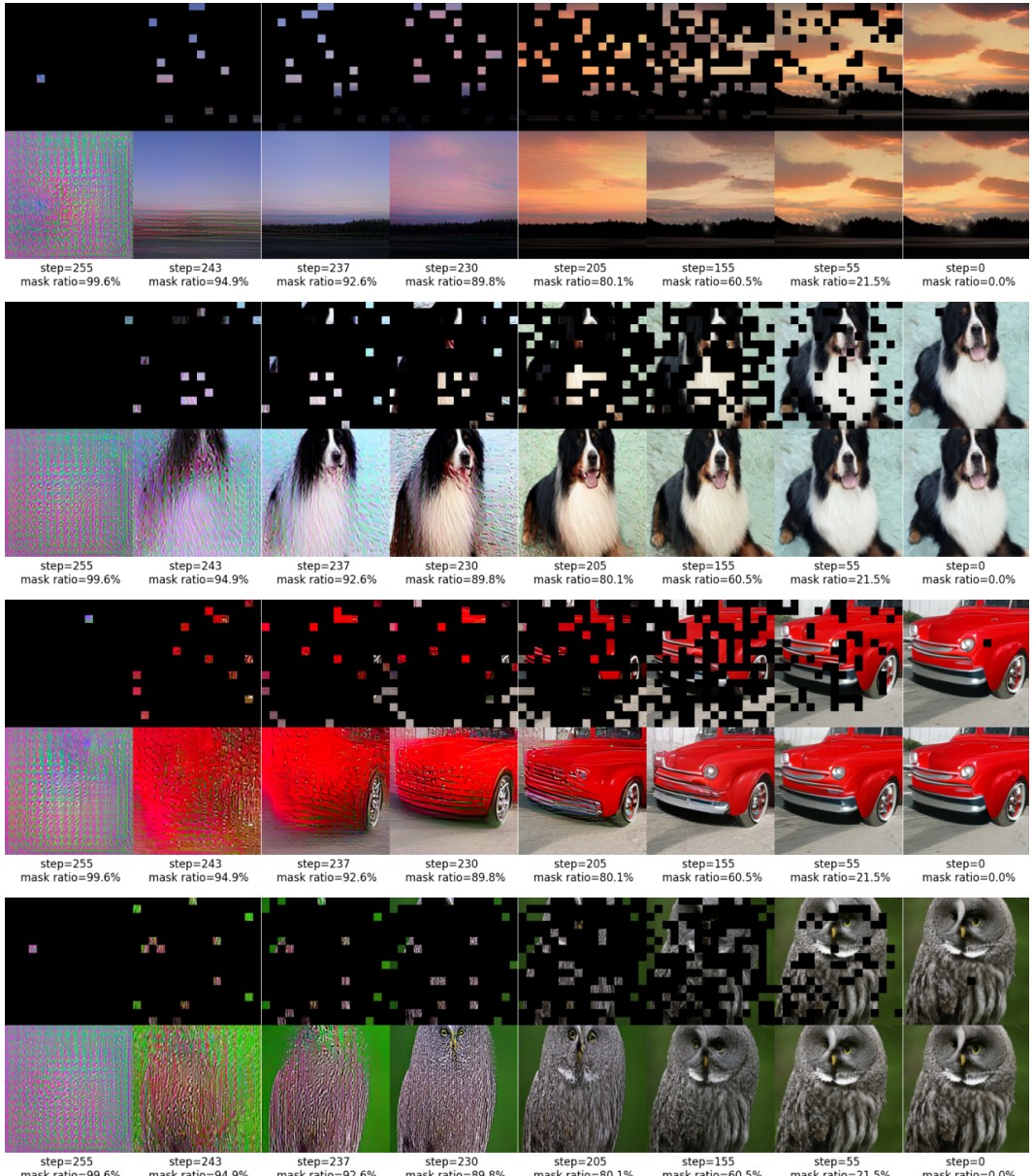

Figure 8: **Progressive generation results on ImageNet-1k**, using *linear* masking schedule with total inference step $T = 256$.

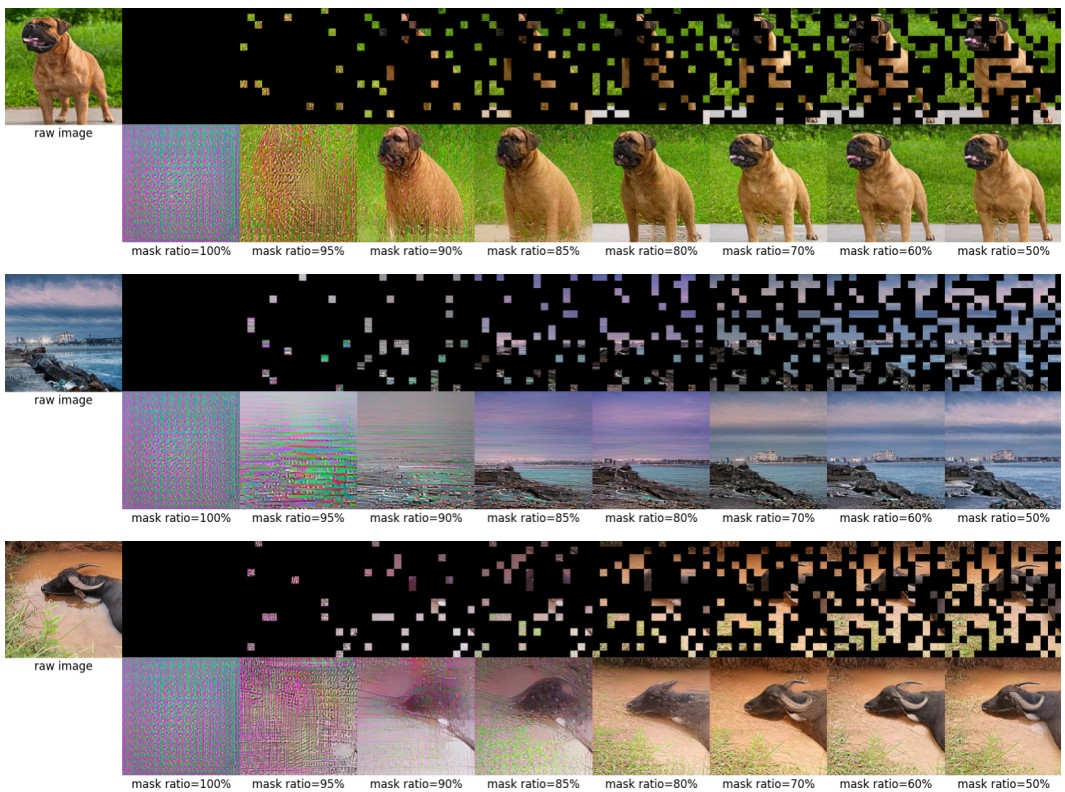

Figure 9: **Synthetic training images with** WeightedSum **strategy.**

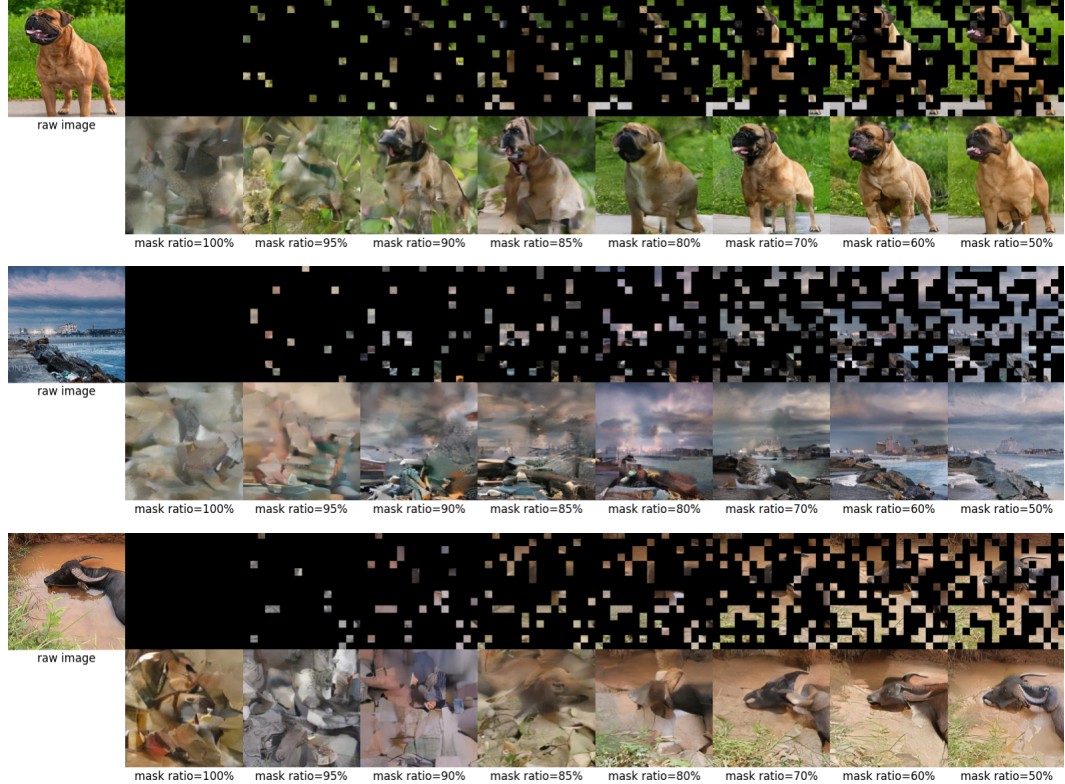

Figure 10: **Synthetic training images with** Sampling **strategy.**

| Hyper-parameters | Value |
| --- | --- |
| Input resolution | $1024 \times 1024$ |
| Finetuning epochs | 100 |
| Warmup length | 250 iters |
| Batch size | 128 |
| Optimizer | AdamW |
| Peak learning rate | $1.6 \times 10^{-4}$ |
| Learning rate schedule | cosine |
| Weight decay | 0.1 |
| Adam $\beta$ | (0.9, 0.999) |
| Layer-wise learning rate decay | 0.8 (B) / 0.9 (L) |
| Augmentation | large scale jittor |
| Stochastic depth | 0.1 (B) / 0.4 (L) |
| Relative positional embeddings | ✓ |

Table 18: **Hyper-parameters for COCO detection.**

| Hyper-parameters | Value |
| --- | --- |
| Input resolution | $512 \times 512$ |
| Finetuning length | 80k iters (B) / 40k iters (L) |
| Warmup length | 1500 iters |
| Batch size | 32 |
| Optimizer | AdamW |
| Peak learning rate | $2 \times 10^{-4}$ (B) / $3.2 \times 10^{-4}$ (L) |
| Learning rate schedule | cosine |
| Weight decay | 0.05 |
| Adam $\beta$ | (0.9, 0.999) |
| Layer-wise learning rate decay | 0.85 |
| Stochastic depth | 0.1 |
| Relative positional embeddings | ✓ |

Table 19: **Hyper-parameters for ADE20k segmentation.**

**Image Inpainting and Outpainting.** ADDP is able to conduct image inpainting and outpainting without further finetuning. Given a masked image, we first generate the initial image $x_t$ by filling the masked region with the average pixels of visible areas. Then the mask ratio and the corresponding timestep $t$ is calculated based on the ratio of the masked area to the entire image. We also use VQ tokenizer to encode $x_t$ into VQ tokens $z_t$. After that, ADDP can generate the final image by continuing the subsequent alternating denoising process. The final output is composited with the input image via linear blending based on the mask, following MaskGIT (Chang et al., 2022). Some results of image inpainting, outpainting and uncropping (outpainting on a large mask) are shown in Fig. 13, Fig. 14 and Fig. 15.

**Intermediate Generated Results with Token Space Visualization.** In Fig. 16, we map the top 3 PCA components of each token embedding (both reliable and unreliable) to RGB values at each intermediate step. Such visualization illustrates that reliable tokens help enhance image sharpness and add more fine-grained details, while unreliable tokens contribute to the refinement of coarse-grained spatial contours. Collectively, these tokens maintain the spatial consistency of the generated image.

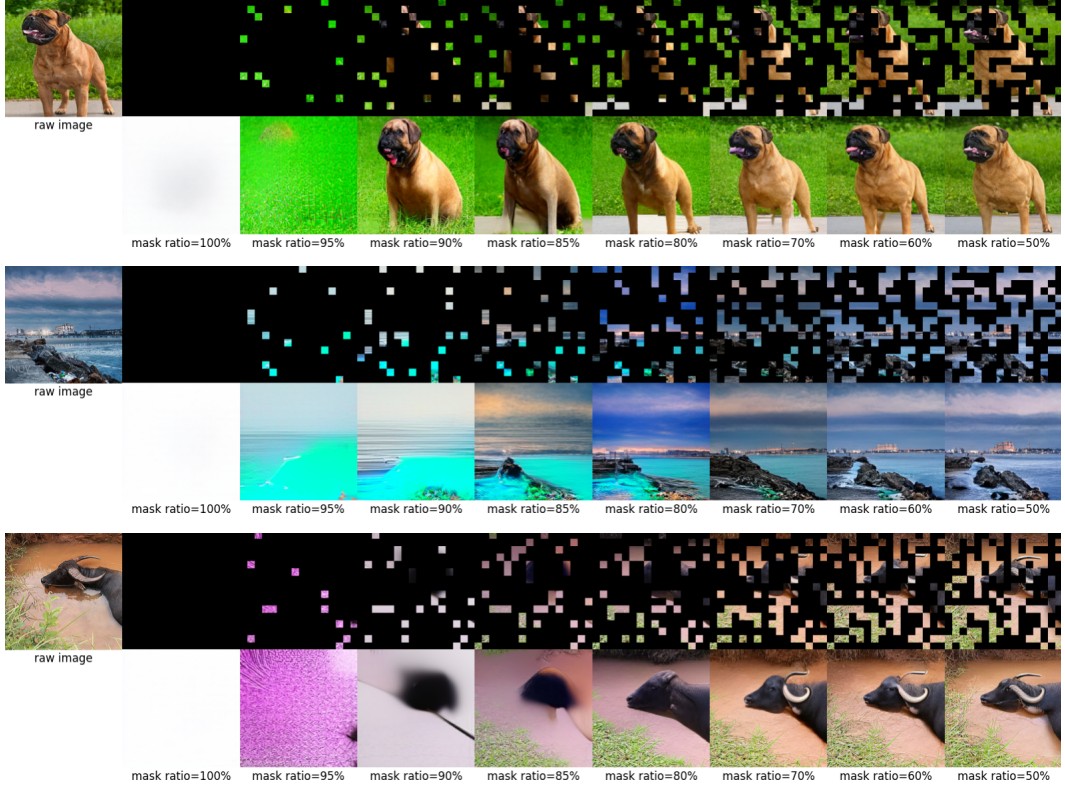

Figure 11: **Synthetic training images with** Argmax **strategy.**

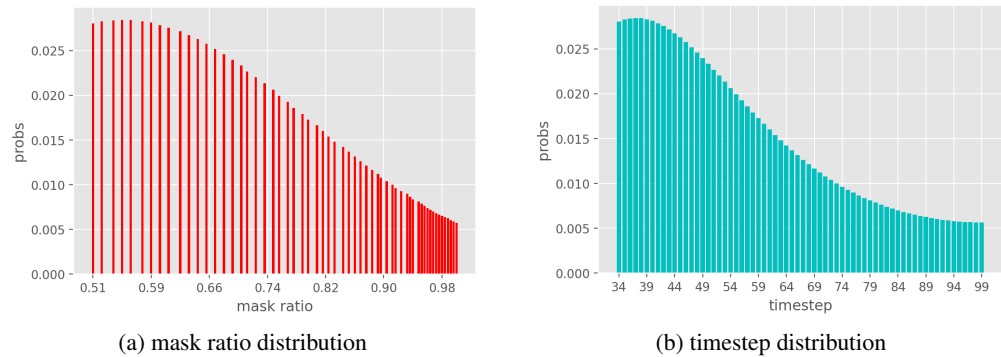

(a) mask ratio distribution          (b) timestep distribution

Figure 12: **The discrete probability distribution used for pre-training.**

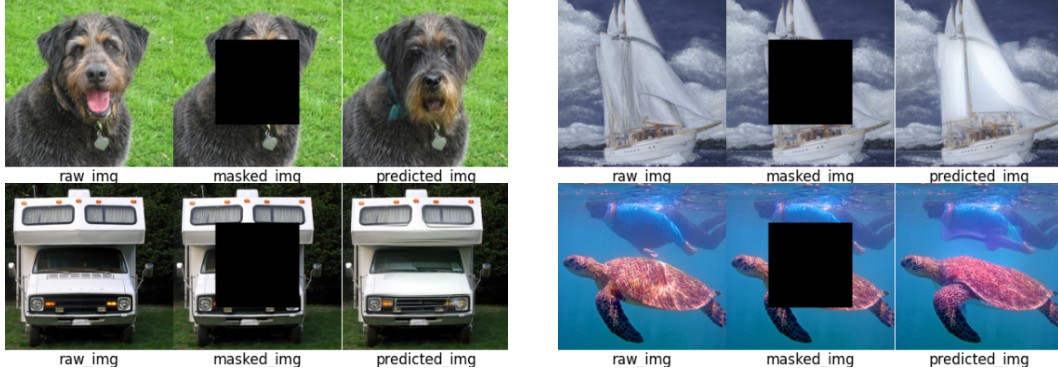

Figure 13: **Results of image inpainting.**

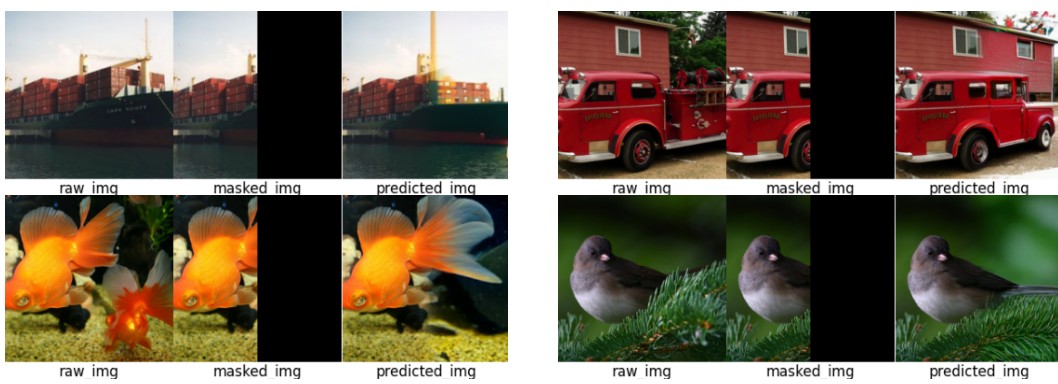

Figure 14: **Results of image outpainting.**

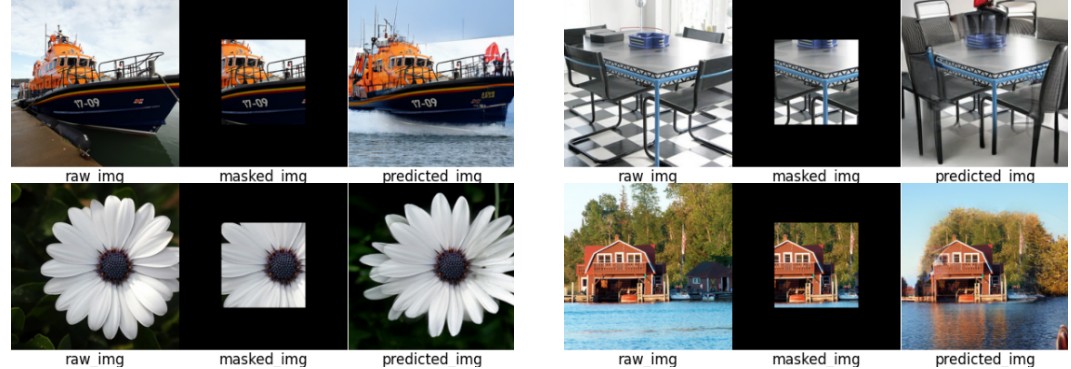

Figure 15: **Results of image uncropping (outpainting on a large mask).**

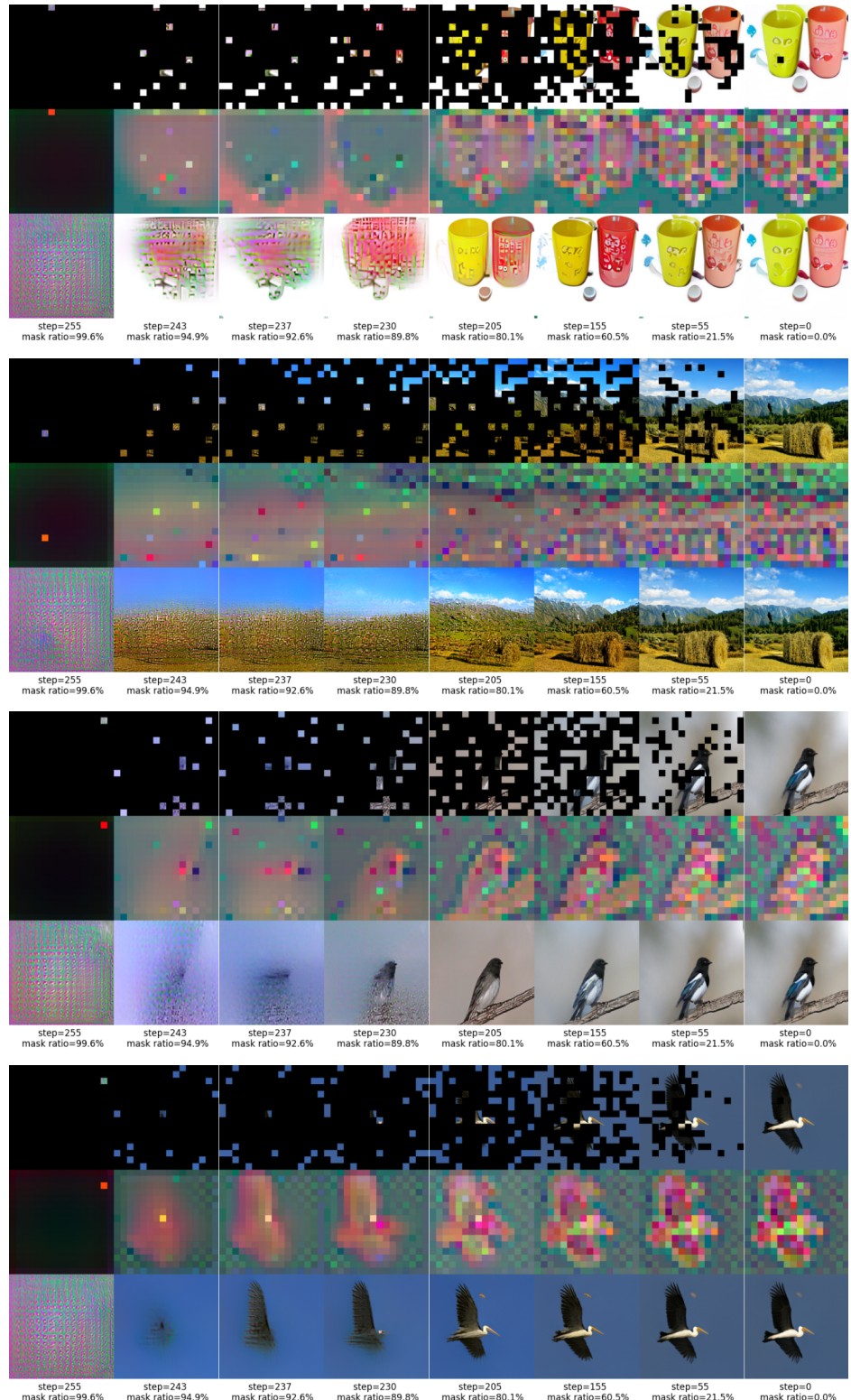

Figure 16: **More progressive generation results on ImageNet-1k**, using *linear* masking schedule with total inference step $T = 256$. For each generated image, the unmasked regions in the top row correspond to the reliable tokens, while the rest corresponds to unreliable tokens at each step. In the second row, the top 3 PCA components of each token embedding (both reliable and unreliable) are mapped to RGB values for visualization. The last row is the result of the noisy synthesized image after Token-to-Pixel Decoding.

APPENDIX REFERENCES

Mahmoud Assran, Mathilde Caron, Ishan Misra, Piotr Bojanowski, Florian Bordes, Pascal Vincent, Armand Joulin, Mike Rabbat, and Nicolas Ballas. Masked siamese networks for label-efficient learning. In *ECCV*, 2022.

Mathilde Caron, Ishan Misra, Julien Mairal, Priya Goyal, Piotr Bojanowski, and Armand Joulin. Unsupervised learning of visual features by contrasting cluster assignments. *NeurIPS*, 2020.

Xinlei Chen and Kaiming He. Exploring simple siamese representation learning. In *CVPR*, 2021.

Jean-Bastien Grill, Florian Strub, Florent Altché, Corentin Tallec, Pierre Richemond, Elena Buchatskaya, Carl Doersch, Bernardo Avila Pires, Zhaohan Guo, Mohammad Gheshlaghi Azar, et al. Bootstrap your own latent-a new approach to self-supervised learning. *NeurIPS*, 2020.

Kaiming He, Georgia Gkioxari, Piotr Dollár, and Ross Girshick. Mask r-cnn. In *ICCV*, 2017.

Dan Hendrycks and Thomas Dietterich. Benchmarking neural network robustness to common corruptions and perturbations. *arXiv preprint arXiv:1903.12261*, 2019.

Dan Hendrycks, Steven Basart, Norman Mu, Saurav Kadavath, Frank Wang, Evan Dorundo, Rahul Desai, Tyler Zhu, Samyak Parajuli, Mike Guo, et al. The many faces of robustness: A critical analysis of out-of-distribution generalization. In *ICCV*, 2021a.

Dan Hendrycks, Kevin Zhao, Steven Basart, Jacob Steinhardt, and Dawn Song. Natural adversarial examples. In *CVPR*, 2021b.

Ari Holtzman, Jan Buys, Li Du, Maxwell Forbes, and Yejin Choi. The curious case of neural text degeneration. In *International Conference on Learning Representations*, 2019.

Yanghao Li, Hanzi Mao, Ross Girshick, and Kaiming He. Exploring plain vision transformer backbones for object detection. In *ECCV*, 2022.

Haohan Wang, Songwei Ge, Zachary Lipton, and Eric P Xing. Learning robust global representations by penalizing local predictive power. *NeurIPS*, 2019.

Ross Wightman, Hugo Touvron, and Hervé Jégou. Resnet strikes back: An improved training procedure in timm. *arXiv preprint arXiv:2110.00476*, 2021.

Tete Xiao, Yingcheng Liu, Bolei Zhou, Yuning Jiang, and Jian Sun. Unified perceptual parsing for scene understanding. In *ECCV*, 2018.

