# OpenReview forum: "ADDP: Learning General Representations for Image Recognition and Generation with Alternating Denoising Diffusion Process"
_ICLR.cc/2024/Conference — ICLR 2024 poster_

### Official Review · Reviewer_1FNr · 2023-10-29

**Soundness:** 3 good
**Presentation:** 3 good
**Contribution:** 3 good
**Rating:** 6
**Confidence:** 3

**Summary:**

The article proposes a novel approach called Alternating Denoising Diffusion Process (ADDP) for general-purpose representation learning in both discriminative and generative tasks. This method combines pixels and the Vector Quantization (VQ) space, alternating between the pixel space and the VQ space during the denoising process. It can generate diverse and high-fidelity images while achieving excellent performance in image recognition tasks. The authors validate the performance of this method in various tasks such as unconditional generation, ImageNet classification, COCO detection, and ADE20k segmentation.

**Strengths:**

1. The article proposes a novel method for learning image representations that can be applied to both image generation and recognition tasks.
2. The proposed method achieves promising results in unconditional image generation and image recognition tasks.
3. Detailed ablation experiments are conducted to verify the effectiveness of each introduced module.

**Weaknesses:**

1. $z_{t-1}$ and $\bar{z}_{t-1}$ are conditionally independent, but why does using $z_t$ to predict $\bar{z}_{t-1}$ result in significantly better performance?
2. The performance of the generative models on ImageNet-256 in Table 2 is no longer state-of-the-art. Updated results of recent image generation models need to be included.
3. Due to the encoder being trained on noisy images, there is a significant drop in performance in Linear Probing.
4. Does expanding the training dataset to include both original images and noisy images improve image recognition performance?
5. Based on my understanding, VQ space is used for image generation while the output space of the encoder is used for image recognition. Are these two spaces independent, and is it possible to merge them into the same space? If not, is it feasible to directly fine-tune models based on the VQ space for image recognition? How does it perform?
6. The illustrated masked tokens $\bar{z}_{t-1}$ and $z_{t-1}^{pred}$ in Figure 4 are not matching.

**Questions:**

See weaknesses.

---

> ### Author Response · Authors · 2023-11-22
> **Response to Reviewer 1FNr (1/2)**
>
> We sincerely appreciate Reviewer 1FNr for the valuable time and constructive suggestions. Some of the suggestions are quite beneficial and may further enhance our method. Please see our detailed response below.
>
> ---
>
> > **Q1: $z_{t-1}$ and $\bar{z}\_{t-1}$ are conditionally independent, but why does using $z_t$ to predict $\bar{z}\_{t-1}$ result in significantly better performance?**
>
> A1:
> There may be a misunderstanding on the experiment setting. Specifically, according to Eq.(6) in the paper, if we change $z\_t$ to $z\_{t-1}$ in the condition, the input image will be generated from $z\_{t-1}$ and $\bar{z}\_{t-1}$ for the consistency of timestep. As a result, the input inadvertently incorporates the essential information of the training targets. Therefore, the main reason for the poor generation performance stems from the leakage of information when using $z\_{t-1}$ to predict $\bar{z}\_{t-1}$.
>
> To avoid confusion, we have revised our paper to remove the statement of conditionally independent.
>
> ---
>
> > **Q2: The performance of the generative models on ImageNet-256 in Table 2 is no longer state-of-the-art. Updated results of recent image generation models need to be included.**
>
> A2:
> We need to clarify that we report the results of **unconditional generation** on ImageNet-256. To the best of our knowledge, the previous state-of-the-art on this benchmark is MAGE[1], which we have already included in the comparison. It is possible that there are omissions in our literature review. We are happy to include more comparisons if the reviewer can suggest some relevant works.
>
> ---
>
> > **Q3: Due to the encoder being trained on noisy images, there is a significant drop in performance in Linear Probing.**
>
> A3:
> First, it is important to emphasize that the core capability of self-supervised learning (SSL) lies in fine-tuning, not linear-probing. Poor linear-probing should not negate the models. MAE [2] and BEiT [3] have highlighted this point. Still, we recognize that this is a potential direction for future research, and we will take this aspect into consideration in future work.
>
> Nevertheless, we explored the reasons for our poor linear-probing. As discussed in Sec. 4.2, our model is trained on noisy synthetic images, making it hard in linear-probing on real images. We speculate that training on real images may help improve the linear-probing performance. To validate this, we fine-tuned our final model for 50 epochs with a 50% input mask ratio and z0 as targets. The linear-probing result improved from 11.5% to 41.2% (better than BEiT), indicating that using data closer to real images indeed enhances linear-probing performance.
>
> ---
>
> > **Q4: Does expanding the training dataset to include both original images and noisy images improve image recognition performance?**
>
> A4:
> Thanks for your suggestion. We tried to mix original images and noisy images during training with a mixing ratio of 1:1. We list the results below. It shows that incorporating original images can improve image recognition performance. Additionally, we observe that incorporating original images will also cause a decline in the quality of image generation. This may be due to the inconsistency of image distribution between training and inference, and the mixing ratio of 1:1 may not be appropriate. The optimal mixing ratio may need further study to achieve a balance between recognition and generation performances.
> | Training Data | FID $\downarrow$ | IS $\uparrow$ | FT Top1 acc $\uparrow$ |
> | --- | --- | --- | --- |
> | noisy images | 23.6 | 33.1 | 83.5 |
> | original:noisy=1:1 | 214.6 | 3.1 | 83.6 |

---

> ### Author Response · Authors · 2023-11-22
> **Response to Reviewer 1FNr (2/2)**
>
> > **Q5: Based on my understanding, VQ space is used for image generation while the output space of the encoder is used for image recognition. Are these two spaces independent, and is it possible to merge them into the same space? If not, is it feasible to directly fine-tune models based on the VQ space for image recognition? How does it perform?**
>
> A5:
> Your interpretation is correct. The output space of the encoder is used for image recognition. The decoder will further transform the encoder output space to the VQ space for image generation. These two spaces are interconnected via the decoder.
>
> Based on your experiment suggestion, we find it possible to merge them into the same space. Specifically, we conducted an experiment where we directly fine-tune the model by using the representations from the decoder output space for image recognition. The results are listed below. It's shown that fine-tuning with representations from the decoder output space can deliver strong image recognition performance. While some gain may be attributed to the increased parameters of the decoder, it indicates the feasibility to merge the space for recognition and generation. We will try to explore this potential direction, which may further simplify the pipeline.
> | Fine-tuned Model | Representation Space | Top1 acc $\uparrow$ |
> | --- | --- | --- |
> | Encoder | Encoder output space | 83.9 |
> | Encoder+Decoder | Decoder output space | 84.5 |
>
> ---
>
> > **Q6: The illustrated masked tokens $\bar{z}\_{t-1}$ and $z\_{t-1}^{pred}$ in Figure 4 are not matching.**
>
> A6: Thanks for your careful reading. We have fixed it in the revised version.
>
> [1] Tianhong Li, Huiwen Chang, Shlok Kumar Mishra, Han Zhang, Dina Katabi, and Dilip Krishnan. Mage: Masked generative encoder to unify representation learning and image synthesis. In CVPR. 2023
>
> [2] Kaiming He, Xinlei Chen, Saining Xie, Yanghao Li, Piotr Dollar, and Ross Girshick. Masked Autoencoders Are Scalable Vision Learners. In CVPR, 2022.
>
> [3] Hangbo Bao, Li Dong, Songhao Piao, and Furu Wei. BEiT: BERT Pre-Training of Image Transformers. In ICLR, 2022.

---

### Official Review · Reviewer_hM7K · 2023-10-31

**Soundness:** 2 fair
**Presentation:** 2 fair
**Contribution:** 2 fair
**Rating:** 5
**Confidence:** 3

**Summary:**

This paper presents an alternating process that simultaneously generates tokens and pixels. This pipeline is suitable for both generation and recognition tasks. However, the additional encoder's effectiveness is unclear as it lacks clear motivation and results. In recognition tasks, the encoder is only applied to pure images, making its training on noisy images relatively meanless.

**Strengths:**

This paper investigates the feasibility of simultaneously generating pixels and tokens to design a university representation for both generation and recognition tasks.

**Weaknesses:**

Although this pipeline yields relatively good results for both generation and recognition tasks, the issue lies in its redundancy and lack of relevance. The additional encoder does not appear to significantly contribute to the generation task, as token-to-pixel and pixel-to-token iterations do not enhance generation. Rather, the additional encoder is solely utilized for recognition purposes. Essentially, this pipeline merely combines two effective submodels without discovering mutual benefits.

**Questions:**

What is the role of token-to-pixel and pixel-to-token at each step of the generation task? Perhaps performing it only once at the end would yield similar results in generation.

---

> ### Author Response · Authors · 2023-11-22
> **Response to Reviewer hM7K**
>
> We thank Reviewer hM7K for the comments and suggetions. To clarify, we aim to elaborate on the motivation and corresponding experimental results pertaining to our incorporation of an additional pixel-based image encoder for both recognition and generation tasks.
>
> ---
>
> > **Q1: Training on noisy images is relatively meaningless for recognition tasks.**
>
> A1:
> No, it is not meaningless. This is a misunderstanding for representation learning. Training on noisy images is a recent mainstream paradigm of representation learning for recognition tasks. A lot of previous methods[1,2,3] have demonstrated that pretraining on noisy images can greatly boost the recognition performances on downstream tasks. In particular, CIM[3] is trained on corrupted images generated by VQGAN, and shows superior performances on various benchmarks. ADDP shares similar spirits on recognition tasks with these works. The results in Tab. 2 also demonstrate the effectiveness of our work.
>
> ---
>
> > **Q2: The additional encoder does not appear to significantly contribute to the generation task, as token-to-pixel and pixel-to-token iterations do not enhance generation. Rather, the additional encoder is solely utilized for recognition purposes. Essentially, this pipeline merely combines two effective submodels without discovering mutual benefits.**
>
> A2:
> No, we first clarify that token-to-pixel and pixel-to-token iteration does help enhance the generation performance. Without them, our process will degenerate to a token-to-token only process, which is essentially MAGE[4]. Under fair comparison, we achieve significantly better performance on image generation (FID 7.6 vs 9.1), improving the performance by 16.5%.
>
> Specifically, MAGE[4] and ADDP have the similar encoder-decoder structure and number of trainable parameters. The key difference is we additionally introduce the pixel images in the generation process. This enables our model to perform denoising based on images rendered by VQ Decoder, instead of relying solely on discrete tokens in the latent space. These rendered images not only retain the information encoded in the tokens, but also introduce the token-to-pixel mapping information stored in the VQ Decoder. This additional information offers more accurate support for the denoising process.
>
> Moreover, as suggested by Reviewer 1FNr, we find that actually the representation space for recognition and generation can also be unified, without any changes to our method. Specifically, we conducted an experiment where we directly fine-tune the learned model by using the representations from the decoder output space for image recognition. As is shown in the table below, it achieves even better performance than our default setting (i.e. finetuning the encoder only).
>
> These results demonstrate the mutual benefits between the encoder and the decoder for both recognition and generation tasks.
> | Fine-tuned Model | Representation Space | Top1 acc $\uparrow$ |
> | --- | --- | --- |
> | Encoder | Encoder output space | 83.9 |
> | Encoder+Decoder | Decoder output space | 84.5 |
>
> ---
>
> > **Q3: What is the role of token-to-pixel and pixel-to-token at each step of the generation task? Perhaps performing it only once at the end would yield similar results in generation.**
>
> A3:
> As mentioned above, if we solely perform token-to-token in the denoising process and token-to-pixel only once at the end, ADDP will degenerate to MAGE[4], whose generation performance is inferior to ours. The token-to-pixel and pixel-to-token iterations enable our model to make better use of the information within the VQ-Decoder at each denoising step, rather than only at the final step.
>
> [1] Kaiming He, Xinlei Chen, Saining Xie, Yanghao Li, Piotr Dollar, and Ross Girshick. Masked Autoencoders Are Scalable Vision Learners. In CVPR, 2022.
>
> [2] Hangbo Bao, Li Dong, Songhao Piao, and Furu Wei. BEiT: BERT Pre-Training of Image Transformers. In ICLR, 2022.
>
> [3] Yuxin Fang, Li Dong, Hangbo Bao, Xinggang Wang, and Furu Wei. Corrupted Image Modeling for Self-Supervised Visual Pre-Training. In ICLR, 2023.
>
> [4] Tianhong Li, Huiwen Chang, Shlok Kumar Mishra, Han Zhang, Dina Katabi, and Dilip Krishnan. Mage: Masked generative encoder to unify representation learning and image synthesis. In CVPR, 2023.

---

### Official Review · Reviewer_pb3B · 2023-10-31

**Soundness:** 2 fair
**Presentation:** 1 poor
**Contribution:** 3 good
**Rating:** 6
**Confidence:** 2

**Summary:**

The paper "ADDP: Learning General Representations for Image Recognition and Generation with alternating denoising diffusion process" attempts to construct models which can understand raw pixels while, at the same time, allowing the generation of visual representations. The proposed architecture is evaluated on the ImageNet-1k, COCO and ADE20k data sets. The paper's main contribution is the integration of image classification, segmentation and generation within a single architecture.

**Strengths:**

- Joint representation learning, in this case for image understanding and generation,  is an interesting research problem. The paper's primary research question is a good fit for ICLR.
- The number of cited papers in the related work section is extensive.
- To the best of my knowledge, the main contribution is novel.
- Experimental results are convincing, especially the extension to segmentation tasks on COCO and ADE20k.

**Weaknesses:**

- The text is hard to follow. Additional copy editing may help here.
- The related work seems to focus on listing many papers. It would help if the related work would attempt to explain some of the key concepts instead of just listing them.
- Figure text is often too small to read in print.

**Questions:**

- What is the structure of the VQ-Decoder in equation 1?
- Are VQ-Tokens defined following van den Oord?
- What is the VQ-Decoder in equation 3? Which one is used?
- What is the encoder structure in equation two?
- Where does the mask in Figure 5(a) come from?
- Why can the MAGE-L network be discarded during inference?
- What do the lock symbols mean in Figure 4? Perhaps the locks signify constant network elements?

---

> ### Author Response · Authors · 2023-11-22
> **Response to Reviewer pb3B (1/2)**
>
> We are grateful for Reviewer pb3B's comments and suggestions. The questions are answered below.
>
> ---
>
> > **Q1: The text is hard to follow. Additional copy editing may help here.**
>
> A1: We apologize for the inconvenience caused. Due to the limited space, we have simplified the background and assumed that the reader has relevant background knowledge. We also note that other reviewers like Reviewer SakC and Reviewer 1FNr have commended the clarity of our paper's presentation. We will be more than willing to provide more details and revise our paper. Below are the answers to other raised questions.
>
> ---
>
> > **Q2: The related work seems to focus on listing many papers. It would help if the related work would attempt to explain some of the key concepts instead of just listing them.**
>
> A2: Due to the limited paper length, it is impractical to exhaustively detail the key concepts of all referenced papers. We assume our readers have a basic understanding of the relevant background knowledge. Therefore, in the related work section, we mainly focus on discussing key points crucial for elucidating the concepts central to our paper.
>
> Specifically, in "Deep Generative Models for Image Generation", Para. 1 aims to introduce the two-stage latent space paradigm proposed by VQ-VAE[1], while Para. 2 discusses the emergence of diffusion models and their integration with VQ-VAE[1].
>
> In "Generative Pre-training for Image Representation Learning", we focus on the development of generative representation learning, particularly for recognition tasks.
>
> The subsequent section, "Generative Modeling for Unifying Representation Learning and Image Generation", examines how previous studies have tackled the challenge of learning general representation for both generation and recognition tasks.
>
> Additionally, at the end of each subsection, we articulate the relationships and distinctions between these approaches and our ADDP framework.
>
> ---
>
> > **Q3: Figure text is often too small to read in print.**
>
> A3: Thanks for the suggestion. The figure texts are enlarged in the revised version.
> In addition, we suggest relocating this comment to the 'Questions' section for appropriate categorization, because issues related to figure text formatting are not typically considered major weaknesses (as per the reviewing [instructions](https://iclr.cc/Conferences/2024/ReviewerGuide#Reviewing%20instructions) and [examples](https://iclr.cc/Conferences/2024/ReviewerGuide#Review%20examples) provided in the ICLR reviewer guidelines).
>
>
> ---
>
> > **Q4: What is the structure of the VQ-Decoder in equation 1?**
>
> A4: As mentioned in "Token-to-Pixel Decoding" of Sec. 3.1, the VQ-Decoder in Eq.(1) can utilize any pre-trained VQ-based model. In our implementation, as discussed in Sec. 4.1, we employ the VQGAN model released by MAGE[2] as the VQ-Decoder here. Basically, it is a convolution-based network that decodes the VQ tokens into raw pixel images.
>
> ---
>
> > **Q5: Are VQ-Tokens defined following van den Oord?**
>
> A5: Yes. The VQ-Tokens in our paper corresponds to what is referred to as “latent variable” in VQ-VAE[1]. We adopt the term "VQ Tokenizer" to denote the VQ Encoder following Muse[3]. Accordingly, "VQ-Tokens" refers to the outputs produced by the VQ Tokenizer.
>
> ---
>
>
> > **Q6: What is the VQ-Decoder in equation 3? Which one is used?**
>
> A6: The VQ-Decoder in Eq.(3) is identical to the one described in Eq.(1). We utilize the VQ-Decoder released by MAGE[2] in our implementation.
>
> ---
>
> > **Q7:What is the encoder structure in equation two?**
>
> A7: Our method imposes no constraints on the structure of the encoder. Theoretically, it can be any model capable of processing images as input and producing representations. In our experiments, we employed ViT-L, ViT-B and ResNet50 as encoders. For detailed results, please refer to Tab. 2, Tab. 7 and Tab. 8 respectively.

---

> ### Author Response · Authors · 2023-11-22
> **Response to Reviewer pb3B (2/2)**
>
> > **Q8: Where does the mask in Figure 5(a) come from?**
>
> A8: The mask in Fig. 5(a) represents the <MASK> token embedding, which is consistent with the <MASK> in Fig. 4. Essentially, it is a randomly initialized trainable tensor.
>
> ---
>
> > **Q9: Why can the MAGE-L network be discarded during inference?**
>
> A9: As detailed in the text prior to Eq.(4), the token predictor (MAGE-L) can be discarded because it is only used to generate $q(\bar{z}_{t-1} | z_t)$, which our model has learnt to predict during training. Consequently, our model inherently incorporates the function of the token predictor during inference.
>
> ---
>
> > **Q10: What do the lock symbols mean in Figure 4? Perhaps the locks signify constant network elements?**
>
> A10: That’s correct. Utilizing the lock symbol to denote the frozen components of the network during training is a common practice. We apologize for not providing this clarification in the original figure caption and have rectified this oversight in the revised version.
>
> [1] Aaron van den Oord, Oriol Vinyals, and Koray Kavukcuoglu. Neural Discrete Representation Learning. In NeurIPS, 2017.
>
> [2] Tianhong Li, Huiwen Chang, Shlok Kumar Mishra, Han Zhang, Dina Katabi, and Dilip Krishnan. Mage: Masked generative encoder to unify representation learning and image synthesis. In CVPR, 2023.
>
> [3] Huiwen Chang, Han Zhang, Jarred Barber, AJ Maschinot, Jose Lezama, Lu Jiang, Ming-Hsuan Yang, Kevin Murphy, William T. Freeman, Michael Rubinstein, Yuanzhen Li, and Dilip Krishnan. Muse: Text-To-Image Generation via Masked Generative Transformers. arXiv preprint arXiv:2301.00704, 2023.

---

### Official Review · Reviewer_SakC · 2023-11-06

**Soundness:** 3 good
**Presentation:** 2 fair
**Contribution:** 3 good
**Rating:** 6
**Confidence:** 4

**Summary:**

Motivated by recent advances in recognition using pixels as inputs and generation via VQ tokens, this paper presents an alternative denoising diffusion process (ADDP) that leverages the strengths of both domains. ADDP utilizes both raw-pixel and VQ spaces to perform recognition and generation tasks. The proposed method includes a Token-to-Pixel decoding stage that generates visual image pixels from VQ tokens. Subsequently, in the Pixel-to-Token Generation stage, the proposed method predicts VQ tokens from noisy images. The process employs an alternative denoising approach that generates pairs of reliable and unreliable tokens before producing noise-free images. The overall architecture is then tested for visual recognition and generation tasks using the ImageNet, COCO, and ADE20K datasets. Moreover, the ablation study of unreliable tokens and the mapping function, as well as prediction targets and masking ratios, demonstrates the effectiveness of the model.

**Strengths:**

-	Overall, the proposed idea of leveraging both token space and raw space appears interesting.
-	The paper's material is presented clearly, and from my perspective, the overall method seems sound.

**Weaknesses:**

-	While the overall concept of the paper is appealing, I believe additional evaluations (as mentioned below) would enhance the paper.

-	I am concerned that the current approach of utilizing VQ-AE for the diffusion process and the token-to-pixel conversion could diminish the generative diversity of the model.

-	I think the paper should acknowledge the limitations of the methods more openly and aim for greater clarity and specificity.

**Questions:**

-	Could you provide more details on the criteria used to generate reliable and unreliable tokens? It is unclear which specific mechanisms within the model determine a token's reliability. An ablation study focusing on this aspect could offer deeper insights into the significance and impact of this feature.

-	How can we ensure that the pixel-to-token generation process does not become constrained by a limited range of samples, especially when utilizing a frozen VQ Encoder-Decoder? Additionally, it would be beneficial to see quantitative comparisons of the proposed method against non-VQ generative models, specifically concerning the diversity of generated samples.

-	Evaluating the dependence of the overall model's performance on the VQ Autoencoder's efficacy would likely yield valuable information about the model's robustness. Consider conducting such an evaluation to provide a clearer understanding of this relationship.

-	The performance gap observed between ADDP and other methodologies in the Linear ImageNet benchmark warrants a comprehensive explanation. Could you elucidate the detailed reasons behind this discrepancy?

-	Finally, visualizing the token spaces through projection to a 2D plane could offer a more intuitive understanding of the model's performance. Have you considered including such visualizations or projections to aid in the evaluation of the model's effectiveness?

---

> ### Author Response · Authors · 2023-11-22
> **Response to Reviewer SakC (1/2)**
>
> We thank Reviewer SakC for the insightful suggestions, and for commenting that the idea is interesting and the paper is presented clearly. These suggestions have been instrumental in deepening our understanding and refining the presentation of our work. Detailed responses to the questions are provided below.
>
> ---
>
> > **Q1: Details on the criteria used to generate reliable and unreliable tokens. (Additional Evaluation) Ablation study on the mechanism to determine reliability.**
>
> A1: Thanks for the suggestions. As outlined in Appendix D.4, we generate reliable tokens using an iterative decoding strategy following MaskGIT[1] during inference.
>
> New reliable tokens are determined based on the predicted probability $p_\theta(\bar{z}_{t-1} | x_t)$ for a given timestep $t$. Specifically, for each masked location, we first sample a candidate token id and its corresponding probability $p$ from the predicted distribution. Then the confidence score $s$ of each location is calculated by adding a Gumbel noise $\epsilon$ to the log probability, i.e. $s = log(p) + \tau \epsilon$. Here, $\tau$ is set to be $6.0 \times \frac{t}{T}$ by default. Tokens with higher confidence scores are selected as the new reliable tokens, while those with lower scores remain to be masked for the next timestep.
>
> During the training phase, without losing generalization, we opt to randomly sample the reliable tokens. The total number of reliable tokens is determined by the sampled timestep $t$. Specifically, the set of reliable tokens at step $t$ invariably includes the reliable tokens at step $t+1$.
>
> We also conduct an additional ablation study to assess the impact of different mechanisms on determining token reliability during inference, as detailed in the tables presented below. Two key ablation choices are considered here:
>
> 1. truncating the probability distribution employed for sampling each candidate token by using nucleus sampling[8], denoted by `top-p`;
> 2. varying the value of $\tau$ to adjust the randomness introduced by the Gumbel noise when computing the confidence scores.
>
> | `top-p` | FID $\downarrow$ | IS $\uparrow$ |
> | --- | --- | --- |
> | 1.0 (default) | 7.6 | 105.1 |
> | 0.95 | 7.9 | 117.3 |
> | 0.9 | 10.5 | 124.1 |
> | 0.7 | 34.1 | 88.6 |
> | 0.5 | 90.9 | 32.9 |
> | 0.3 | 185.8 | 8.4 |
> | 0.1 | 325.8 | 2.3 |
>
> | $\tau$ | FID $\downarrow$ | IS $\uparrow$ |
> | --- | --- | --- |
> | 0.0 | 353.7 | 1.3 |
> | 2.0 | 18.5 | 107.2 |
> | 6.0 | 12.1 | 80.7 |
> | 20.0 | 27.3 | 48.7 |
> | 60.0 | 33.8 | 41.0 |
> | $2.0 \times \frac{t}{T}$ | 23.3 | 109.4 |
> | $6.0 \times \frac{t}{T}$ (default) | 7.6 | 105.1 |
> | $20.0 \times \frac{t}{T}$ | 17.7 | 63.9 |
> | $60.0 \times \frac{t}{T}$ | 27.9 | 48.3 |
>
> Our findings indicate that the current setup is nearly optimal, yielding the lowest FID. However, it's noteworthy that the IS score benefits from slightly reducing the value of `top-p` and $\tau$. This suggests that disregarding tokens with excessively low confidence may enhance the quality of synthesized images.
>
> The discussions and experiments above have been added into the appendix C.4 of our revised version.
>
> ---
>
> > **Q2: How to ensure the pixel-to-token generation process is not limited by a frozen VQ Encoder-Decoder? (Additional Evaluation) Quantitative comparison on the generation diversity.**
>
> A2: Employing a frozen VQ Encoder-Decoder is a well-established practice for image generation, as evidenced by its adoption in previous works such as MAGE[2], MaskGIT[1], and Muse[3]. Therefore, the limitation (if exists) caused by the VQ Encoder-Decoder is widespread, and thus is not the main focus of this paper.
>
> As for the evaluation of generation diversity, the FID used in our paper can capture both diversity and fidelity according to [5], where a lower FID indicates not only improved visual quality but also increased diversity.
>
> To further analyze diversity, we employ precision and recall metrics following ADM[5] as well. Higher recall indicates better diversity. The results are detailed in the table below.
>
> Although the non-VQ model ADM[5] without classifier guidance achieves high diversity, its fidelity is significantly worse than ours. Besides that, our method achieves both better fidelity and diversity compared to ADM[5] with classifier guidance and MAGE[2] on unconditional generation.
>
> We have updated the experiment and discussion into appendix C.5 of our revised version.
>
> | Method | FID $\downarrow$ | IS $\uparrow$ | Precision $\uparrow$ | Recall $\uparrow$ |
> | --- | --- | --- | --- | --- |
> | ADM (w/o classifier guidance) | 26.2 | 39.7 | 0.61 | 0.63 |
> | ADM (w/ classifer guidance) | 12.0 | 95.4 | 0.76 | 0.44 |
> | MAGE | 9.1 | 105.1 | 0.75 | 0.47 |
> | Ours | 7.6 | 105.1 | 0.74 | 0.49 |

---

> ### Author Response · Authors · 2023-11-22
> **Response to Reviewer SakC (2/2)**
>
> > **Q3: (Additional Evaluation) Model's performance dependency on the VQ AutoEncoder.**
>
> A3: It is an interesting and valuable research topic to analyze the dependency between generation model's performance and the pre-trained VQ Autoencoder. However, such topic is orthognal to the core motivation of ADDP, which is to learn general representation for both generation and recognition tasks. We are willing to further explore this direction in our future works.
>
> ---
>
> > **Q4: Detailed reason behind linear probing benchmark.**
>
> A4: First, it is important to emphasize that the core capability of self-supervised learning (SSL) lies in fine-tuning, not linear-probing. Poor linear-probing should not negate the models. MAE [7] and BEiT [6] have highlighted this point. Still, we recognize that this is a potential direction for future research, and we will take this aspect into consideration in future work.
>
> Nevertheless, we explored the reasons for our poor linear-probing. As discussed in Sec. 4.2, our model is trained on noisy synthetic images, making it hard in linear-probing on real images. We speculate that training on real images may help improve the linear-probing performance. To validate this, we fine-tuned our final model for 50 epochs with a 50% input mask ratio and $z_0$ as targets. The linear-probing result improved from 11.5% to 41.2% (better than BEiT), indicating that using data closer to real images indeed enhances linear-probing performance.
>
> ---
>
> > **Q5: (Additional Evaluation) Visualizing the token space.**
>
> A5: In Fig. 9 of the Appendix, we present the visualization of the noisy decoded images at each step, which can be viewed as the projection of token space into raw pixel space.
>
> Additionally, in the revised version of our paper, we also update Fig. 16, which offers direct visualizations of the token space during generation as well. As is shown in Fig. 16, we map the top 3 PCA components of each token embedding (both reliable and unreliable) to RGB values. The visualization intuitively shows that, at each intermediate step, reliable tokens help sharpen the image and add more details, while unreliable tokens help refine the coarse-grained spatial contours. Collectively, these tokens maintain the spatial consistency of the generated image, highlighting the efficacy of our approach.
>
> ---
>
> > **Q6: Acknowledging the limitations of the methods more openly and aiming for greater clarity and specificity.**
>
> A6: We thank reviewer for the suggestion and have accordingly updated the limitations in the revised version. The new version includes a more detailed discussion on the dependency of our method on a pre-trained VQ-Encoder/Decoder. Additionally, it outlines more prospective research directions, such as integrating our method with continuous diffusion process like latent Gaussian diffusion.
>
>
> [1] Huiwen Chang, Han Zhang, Lu Jiang, Ce Liu, and William T. Freeman. MaskGIT: Masked Generative Image Transformer. In CVPR, 2022.
>
> [2] Tianhong Li, Huiwen Chang, Shlok Kumar Mishra, Han Zhang, Dina Katabi, and Dilip Krishnan. Mage: Masked generative encoder to unify representation learning and image synthesis. In CVPR, 2023.
>
> [3] Huiwen Chang, Han Zhang, Jarred Barber, AJ Maschinot, Jose Lezama, Lu Jiang, Ming-Hsuan Yang, Kevin Murphy, William T. Freeman, Michael Rubinstein, Yuanzhen Li, and Dilip Krishnan. Muse: Text-To-Image Generation via Masked Generative Transformers. arXiv preprint arXiv:2301.00704, 2023.
>
> [4] Aaron van den Oord, Oriol Vinyals, and Koray Kavukcuoglu. Neural Discrete Representation Learning. In NeurIPS, 2017.
>
> [5] Prafulla Dhariwal and Alex Nichol. Diffusion Models Beat GANs on Image Synthesis. In NeurIPS, 2021.
>
> [6] Hangbo Bao, Li Dong, Songhao Piao, and Furu Wei. BEiT: BERT Pre-Training of Image Transformers. In ICLR, 2022.
>
> [7] Kaiming He, Xinlei Chen, Saining Xie, Yanghao Li, Piotr Dollar, and Ross Girshick. Masked Autoencoders Are Scalable Vision Learners. In CVPR, 2022.
>
> [8] Holtzman, Ari and Buys, Jan and Du, Li and Forbes, Maxwell and Choi, Yejin. The curious case of neural text degeneration. arXiv preprint arXiv:1904.09751, 2019.

---

### Meta-Review · Area_Chair_gKyH · 2023-12-10

**Metareview:**

The paper presents a method for recognition and generation based on an alternating denoising and diffusion process. Three of the four reviewers thought the paper was above threshold for acceptance. They thought the idea was novel and interesting enough and a good fit for ICLR. The experimental results were convincing. Some concern was raised about the clarity of the presentation, which could be addressed in a final draft.

**Justification For Why Not Higher Score:**

No reviewer argued outright for acceptance

**Justification For Why Not Lower Score:**

The sentiment clearly was for acceptance.

---

### Decision · Program_Chairs · 2024-01-16

Accept (poster)